# How do gravity waves triggered by a typhoon propagate from the troposphere to the upper atmosphere?

Qinzeng Li[1,3], Jiyao Xu[1,2], Hanli Liu[4], Xiao Liu[5], Wei Yuan[1,3]

[1]State Key Laboratory of Space Weather, National Space Science Center, Chinese Academy of Sciences, Beijing,100190, China,

[2]School of Astronomy and Space Science, University of Chinese Academy of Science, Beijing, 100049, China,

[3]Hainan National Field Science Observation and Research Observatory for Space Weather,

[4]High Altitude Observatory, National Center for Atmospheric Research, Boulder, CO 80307-3000, USA,

[5]School of Mathematics and Information Science, Henan Normal University, Xinxiang, 453007, China,

Correspondence to: xujy@nssc.ac.cn

## Abstract

Gravity waves (GWs) strongly affect atmospheric dynamics and photochemistry and the coupling between the troposphere, stratosphere, mesosphere, and thermosphere. In addition, GWs generated by strong disturbances in the troposphere (e.g., thunderstorms and typhoons) can affect the atmosphere of the Earth from the troposphere to the thermosphere. However, the fundamental process of GW propagation from the troposphere to the thermosphere is poorly understood because it is challenging to constrain this process using observations. Moreover, GWs tend to dissipate rapidly in the thermosphere because the molecular diffusion increases exponentially with height. In this study, a double-layer airglow network was used to capture concentric GWs (CGWs) over China that were excited by the Super Typhoon Chaba (2016). We used ERA-5 reanalysis data and Multi-functional Transport Satellite-1R observations to quantitatively describe the propagation processes of typhoon-generated CGWs from the troposphere, through the stratosphere and mesosphere, to the thermosphere. We found that the CGWs in the mesopause region were generated directly by the typhoon in the troposphere. However, the backward ray tracing analysis suggested that CGWs in the thermosphere originated from the secondary waves generated by the dissipation of the CGW and/or nonlinear processes in the mesopause region.

## 1. Introduction

Gravity waves (GWs) can transfer momentum and energy from the lower to the upper atmosphere, thereby affecting global circulation and the thermal and compositional structures in the middle and upper atmospheres (Holton, 1983; Fritts and Alexander, 2003). Studies of dynamical, photochemical, and electrodynamics processes have indicated that GWs are fundamental for the coupling process between the troposphere, stratosphere, mesosphere, and thermosphere (Liu and Vadas, 2013; Smith et al., 2013; Vadas and Liu, 2013; Xu et al., 2015; Vadas and Becker, 2019).

Concentric GWs (CGWs) are a unique type of GW and considered to be mainly generated by convective activity in the troposphere. CGWs can also be generated by GW breaking (Vadas and Becker, 2019; Lund et al., 2020; Kogure et al., 2020) volcanoes (Duncombe, 2022), nuclear explosions (Pfeffer and Zarichny, 1962; Pierceet al.,1971), and rockets (Liu et al., 2020). CGWs in the stratosphere and mesosphere generated by thunderstorms have been widely reported since their sources are ubiquitous (Taylor and Hapgood, 1988; Sentman et al., 2003; Suzuki et al., 2007; Yue et al., 2009; Vadas et al., 2012; Xu et al., 2015; Heale et al., 2019; Smith et al., 2020). In addition, Liu et al. (2014) utilized the Whole Atmosphere Community Climate model to study the global CGWs. In previous studies, CGWs induced by typhoons were detected using ground-based optical remote sensing (Suzuki et al., 2013) while those induced by hurricanes and tropical cyclones were detected using the Suomi National Polar-orbiting Partnership satellite (Yue et al., 2014; Xu et al., 2019) in the mesopause region.

Notably, GWs tend to dissipate rapidly in the upper atmosphere due to molecular

viscosity and thermal diffusion (Vadas, 2007). Thermosphere GWs that are not dissipated
can originate directly from the troposphere (Vadas, 2007; Azeem et al., 2015) or from
secondary GWs, which are generated from the breaking of primary GWs in the
mesosphere or thermosphere region (Vadas and Fritts, 2003; Vadas and Crowley, 2010;
Vadas and Azeem, 2021). Furthermore, Vadas and Becker (2019) for the first time
presented global simulations of tertiary CGWs from the dissipation of secondary CGWs
in the thermosphere. Moreover, wave-wave interaction, wave-mean flow interaction
(Franke and Robinson, 1999; Vadas and Fritts, 2001), self-acceleration, and nonlinear
breaking are other potential secondary wave generation mechanisms (Lund and Fritts,
2012; Fritts et al., 2015; Dong et al., 2020; Fritts et al., 2020; Zhou et al. 2002; Heale et al.
2020). At the same time, tunneling has been deemed as a mechanism that can couple waves
from tropospheric sources to the thermosphere (Walterscheid and Hecht, 2003; Gavrilov
and Kshevetskii; 2018, Heale et al., 2021). However, the lack of observations of the entire
atmosphere limits our understanding of the fundamental process of how GWs propagate
from the lower to the upper atmosphere step by step on the aspect of observations.

This paper presents a case study examining CGWs excited by Super Typhoon

Chaba (2016). To this end, we utilized Multi-functional Transport Satellite-1R
(MTSAT-1R) observations, multi-layer European Centre for Medium-range Weather
Forecasts (ECMWF) ERA-5 reanalysis data (Hoffmann et al., 2019; Hersbach et al., 2020),
and high spatio-temporal resolution double-layer airglow network (DLAN) (Xu et al.,
2021) observations. The CGW observations from the troposphere to the stratosphere and
then to the mesosphere were taken from MTSAT-1R, ERA-5, and the DLAN. However,
given the observational limitations between the mesosphere and thermosphere, the two
layers are connected by ray tracing theory . The objectives of this study were to (a)
investigate  multi-layer CGW features produced by Super Typhoon Chaba (2016) from
near the ground to a height of 250 km, (b) to examine the entire propagation process of the
CGWs excited by typhoon from the lower atmosphere to the upper atmosphere, and (c) to
provide new insights into the coupling between different atmospheric layers.
## 2. Data and Methods
### 2.1 Double layer all-sky airglow imager network data
A DLAN, including an OH layer (~87 km) and OI 630.0 nm layer (~250 km) was
established over mainland China. The research aim of the DLAN is to explore the
physical mechanism of vertical and horizontal propagation and the evolution of
atmospheric waves in the middle and upper atmosphere triggered by severe disasters, such
as typhoons, earthquakes, and tsunamis. The OH airglow network comprises 15 stations,
including the first no-gap OH airglow all-sky imager network located in northern China
(Xu et al., 2015). The OI 630.0 nm airglow network contains 12 stations. Each imager
consists of a $1024 \times 1024$ pixel back-illuminated CCD detector and a Nikon16 mm/2.8D
fish-eye lens with a $180°$ field of view (FOV). The OI 630.0 nm imager is operated at the
3.0 nm bandwidth filter with a central wavelength of 630.0 nm. Observations using
airglow optical remote sensing require only a few airglow imagers to cover a wide area
although it is limited by meteorological conditions. Moreover, airglow observations can be
used to monitor multi-layer GW activities. Figure 1a and 1b illustrate the OH and OI
630.0 nm network station distribution maps, respectively, in China. The OI 630.0 nm
network covers nearly the entire mainland China. Furthermore, the DLAN provides an
excellent solution for studying the coupling processes between the mesosphere and
thermosphere.
Several standard procedures were applied to raw airglow images, including star
contamination subtraction, flat fielding to remove van Rhijin, and atmospheric extinction
(Li et al., 2011). The GW structure was retrieved by taking the deviation of each
processed image from a half-hour running average window image. Finally, the images
were projected onto the Earth's surface using the standard star map software and the
altitude of the airglow layer (Garcia et al., 1997). The altitudes of the OH and OI 630.0
nm emission layers were set as approximately 87 km and 250 km, respectively.

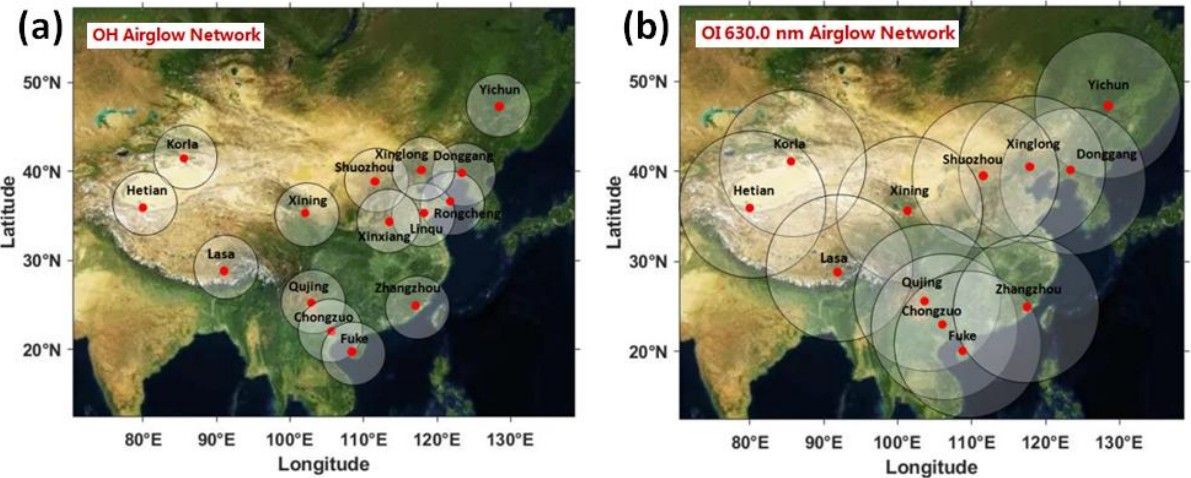


**Figure 1. (a)** OH airglow all-sky imager network (15 stations). **(b)** Red line (630 nm) airglow all-sky
imager network (12 stations). The circles on the maps give the effective observation ranges of OH and
Red line airglow imagers with diameters of about 800 km and 1800 km, respectively.
**2.2 Development of Super Typhoon Chaba**
Super Typhoon Chaba (2016) developed in the north-western Pacific on 24
September 2016 and its track is shown in Fig. 2a. Initially, it moved westward and then
turned north-westward on 30 September. The central pressure in the eye of the typhoon and

the maximum wind speed are shown in Fig. 2b. On 3 October 2016 at 20:00 LT, the

typhoon was in the mature stage with a minimum central pressure of 905 hPa and

maximum sustained winds of approximately 59 m/s. The typhoon moved northward on 4

October 2016 at 02:00 LT until 5 October 2016 at 02:00 LT. The typhoon continued

moving towards the northeast and disappeared on 8 October 2016 at 02:00 LT.

Consecutive satellite images of the typhoon from MTSAT-1R from 18:00 LT on 3 October

2016 to 00:00 LT on 5 October 2016 are shown in Fig. 3. MTSAT-1R, which belongs to the

Japan Meteorological Agency, comprises a series of Geo-stationary Meteorological

Satellites. MTSAT-1R is located at around 140 °E and covers East-Asia and the western

Pacific region. The MTSAT-1R consists of four infrared channels (IR1, IR2, IR3, and IR4)

and one visible channel (VIS). The MTSAT- IR1 was used in this study. The track of the

typhoon was beyond the effective FOV of the OH network and at the edge of the effective

FOV of the OI 630.0 nm network.

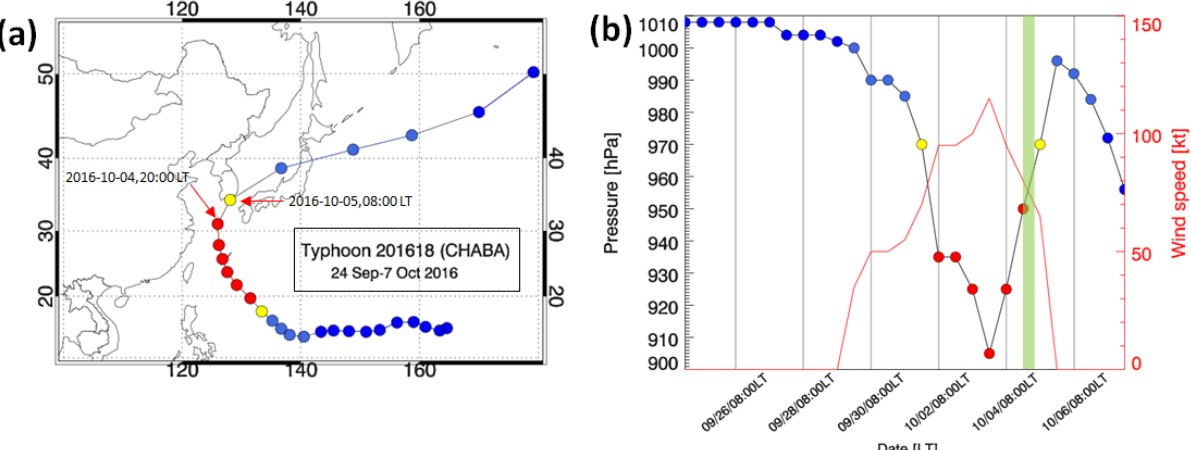

**Figure 2. (a)** The track of Typhoon Chaba is denoted by dots from 24 September to 7 October 2016

every 12 hours. **(b)** Central pressure of Typhoon Chaba corresponding to the tracks in **(a)**. The red line

denotes the maximum sustained wind speed. The green shadow band denotes the time of

ground-based airglow observation from 20:00 LT to 04:00 LT during the night of 4-5 October 2016.

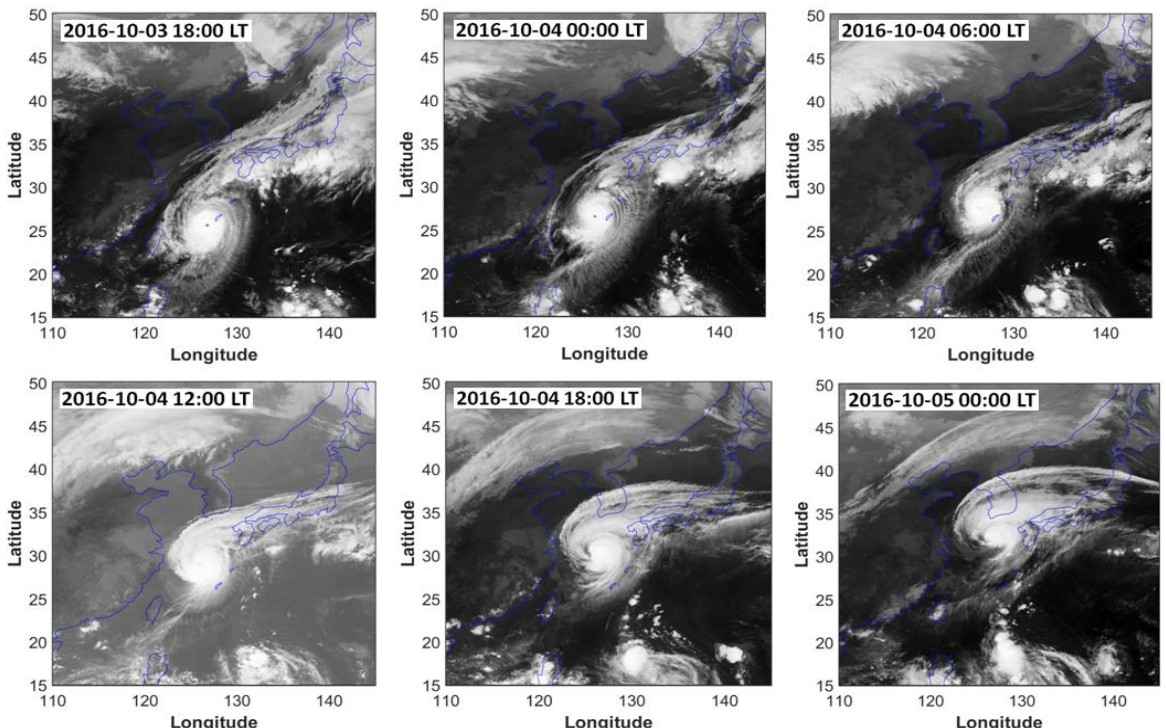

**Figure 3**. Consecutive satellite images of the typhoon Chaba from MTSAT-1R. The period is from 18:00 LT on 3 October 2016 to 00:00 LT on 5 October 2016, with an interval of 6 hours.

**2.3 ERA-5 reanalysis data**

ERA-5 is a fifth-generation ECMWF atmospheric reanalysis that provides hourly data for many atmospheric and wave parameters. ERA-5 is produced using a four-dimensional variational data assimilation algorithm based on Integrated Forecast System (IFS), with 137 hybrid sigma/pressure (model) levels in the vertical from 1000 to 0.01 hPa (0 to 80 km). More details of the model, data assimilation system, and observation data used to produce ERA-5 were described by Hersbach et al. (2020). Horizontal reanalysis temperature and wind data with a pre-interpolated resolution of 0.25 ° × 0.25 ° and time resolution of 1 h were used in this study.

**2.4 Ray tracing model**

We used a ray-tracing method to estimate the source location of the thermospheric

secondary CGWs. The model was based on a dispersion relation that considers molecular
viscosity and thermal diffusivity (Vadas, 2007), as shown in Equation (1):
$$m^2 = \frac{k_H^2 N^2}{\omega_{Ir}^2(1+\delta_+ + \delta^2/\mathrm{Pr})}\left[1+\frac{v^2}{4\omega_{Ir}^2}(\mathbf{k}^2 - \frac{1}{4H^2})^2 \frac{(1-\mathrm{Pr}^{-1})^2}{(1+\delta_+/2)^2}\right]^{-1} - k_H^2 - \frac{1}{4H^2} \quad,$$ (1)
where $\omega_{Ir} = \omega_r - (ku+lv)$ is the intrinsic frequency ($\omega_r$ is ground-based frequency); $\mathbf{k}^2 =$
$k_H^2 + m^2$, $k_H^2 = k^2 + l^2$; $H$ is the scale height; $v = \mu/\bar{\rho}$ is the kinematic viscosity where $\mu$ is the
molecular viscosity and $\bar{\rho}$ is the background density; $\delta = vm/H\omega_{Ir}$, $\delta_+ = \delta(1 + Pr^{-1})$, where
$Pr$ is the Prandtlnumber. $k$, $l$, and $m$ are the zonal, meridional, and vertical wave number
components of the GW, respectively. The horizontal wavelength ($k_H$) of the CGW was
obtained from the ground-based airglow observations; $N^2 = (g/T)(dT/dz + g/c_p)$ is the
square of the Brunt-Väisälä frequency, where $g$ is the gravitational acceleration, $T$ is the
background temperature, $c_p$ is the specific heat at constant pressure. The background
temperature $T$ and density $\bar{\rho}$ were obtained from the NRLMSISE-00 model (Picone et al.,
2002).The group velocity of the wave packet is formalized by Equation (2):
$$c_{gi} = dx_i/dt = \partial\omega_{Ir}/\partial k_i + V_i \quad,$$ (2)
where $V_i (u, v, w)$ is the background wind, which was obtained from the Horizontal Wind
Model 14 (Drob et al., 2015) and $w$ is the vertical wind velocity, which was neglected. In
this study, we assume that the background wind field is independent of time, so
ground-based frequency $\omega_r$ remains constant along a ray's path (Lighthill, 1978).
However, the actual wind field changes with time, which may lead to deviation between
the ray tracing results and the wave source locations.
Using Equations (1)-(2), we yield the ground-based (zonal, meridional, and vertical)
group velocity equation as follows (Vadas and Fritts, 2005):

$$c_{gx} = \frac{k}{\omega_{Ir}\mathrm{B}}\left[\frac{N^2(m^2+1/4H^2)}{(\mathrm{k}^2+1/4H^2)^2} - \frac{v^2}{2}\left(1-\mathrm{Pr}^{-1}\right)^2(\mathrm{k}^2-\frac{1}{4H^2})\frac{(1+\delta_+ + \delta^2/\mathrm{Pr})}{\left(1+\delta_+/2\right)^2}\right] + u, \tag{3}$$

$$c_{gy} = \frac{l}{\omega_{Ir}\mathrm{B}}\left[\frac{N^2(m^2+1/4H^2)}{(\mathrm{k}^2+1/4H^2)^2} - \frac{v^2}{2}\left(1-\mathrm{Pr}^{-1}\right)^2(\mathrm{k}^2-\frac{1}{4H^2})\frac{(1+\delta_+ + \delta^2/\mathrm{Pr})}{\left(1+\delta_+/2\right)^2}\right] + v, \tag{4}$$

$$c_{gz} = \frac{1}{\omega_{Ir}\mathrm{B}}\left\{m\left[-\frac{k_H^2 N^2}{(\mathrm{k}^2+1/4H^2)^2} - \frac{v^2}{2}\left(1-\mathrm{Pr}^{-1}\right)^2(\mathrm{k}^2-\frac{1}{4H^2})\frac{(1+\delta_+ + \delta^2/\mathrm{Pr})}{\left(1+\delta_+/2\right)^2}\right.\right.$$
$$\left.\left.+\frac{v^4\left(1-\mathrm{Pr}^{-1}\right)^4}{16H^2\omega_{Ir}^2}\frac{(\mathrm{k}^2-1/4H^2)^2}{\left(1+\delta_+/2\right)^3} - \frac{v^2}{\mathrm{Pr}\,H^2}\right] - \frac{v_+\omega_{Ir}}{2H}\right\}, \tag{5}$$

where $\mathrm{B} = \left[1 + \dfrac{\delta_+}{2} + \dfrac{\delta^2 v^2}{16\omega_{Ir}^2}\left(1-\mathrm{Pr}^{-1}\right)^4\dfrac{(\mathrm{k}^2-1/4H^2)^2}{\left(1+\delta_+/2\right)^3}\right], \nu_+ = \nu(1+Pr^{-1}).$

## 3. Results

### 3.1 Propagation of typhoon-induced CGWs in the stratosphere

We extracted the stratospheric CGW excited by the typhoon from ERA-5 reanalysis.
Figure 4a, 4b, and 4c show the multilayer temperature perturbations at approximately 60
km ,40 km, and 20 km at 23:00 LT, retrieved from the ERA-5 reanalysis on 4 October 2016,
respectively. Temperature perturbations were calculated by subtracting the background
with a 7 ×7 grid point running mean at 20 km and 17 ×17 grid point running mean at 40
km and 60 km. We found that the temperature disturbance was about ±1.5-2 K at 20 km
and ±3-4 K at 40 km. Using the ECMWF reanalysis data, Kim et al. (2009) reported a
similar temperature disturbance(±4 K) at 40 km altitude. Becker et al. (2022) showed
that typical temperature perturbation amplitudes simulated by a High Altitude
Mechanistic general Circulation Model were ±1-2 K in the wintertime lower stratosphere
and ±5 K in the stratopause region. However, the temperature disturbance at 60 km in
ERA-5 was only ±1.3 K and did not increase with increasing altitude, which may be
caused by this altitude being well within the sponge layer of the reanalysis model. Figure
4d, 4e, and 4f show the corresponding wavelet analysis contours of the red line in Fig. 4a,
4b, and 4c. The expansion area of CGW at the height of 20 km (Fig. 4c) was small, and the
horizontal wavelength was approximately 150 km from Fig. 4f. The CGWs were present
over a large area of 0 °N -50 °N and 100 °E -150 °E at approximately 60 km. The distance of
the CGWs, extending from the center of the circle ranged from 500 km (at approximately
20 km height) to 3000 km (at approximately 60 km height), which suggests that the
larger-scale CGW arrive earlier at higher altitudes (have faster vertical group velocities)
than the smaller-scale waves (Vadas and Azeem, 2021). The ERA-5 reanalysis data was
utilized for characterizing the scale of the CGWs and indicated no small-scale fluctuation.
According to the wavelet analysis of Fig. 4d and 4e, the horizontal wavelengths of the
northward propagating CGW at 60 km (Fig. 4a) and 40 km (Fig. 4b) were approximately
265 km and 290 km, respectively.

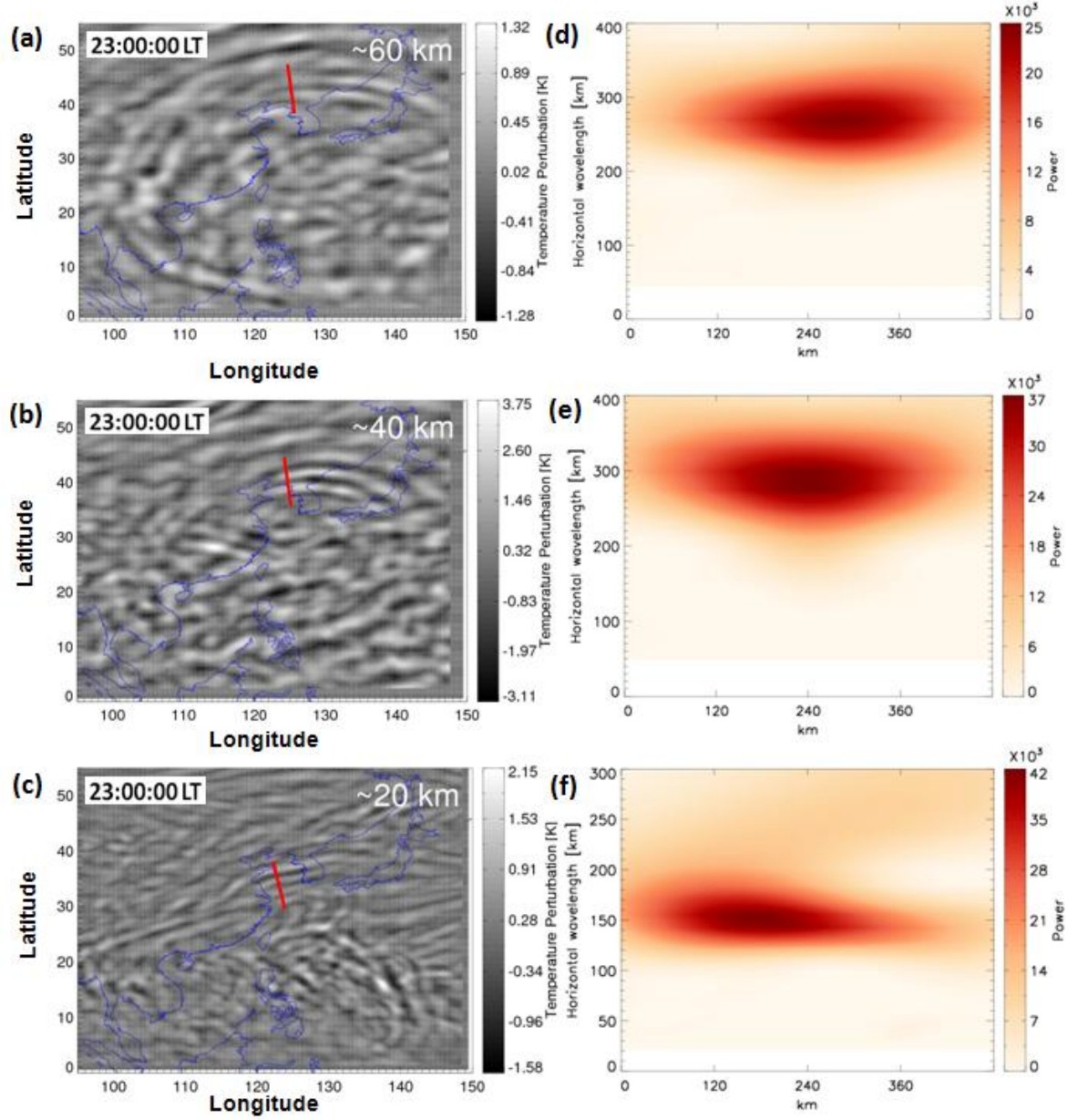

**Figure 4.** Temperature perturbations at **(a)** ~60 km , **(b)** ~40 km, and **(c)**~20 km at 23:00 LT on 4

October 2016 derived from ERA-5 reanalysis. **(d)** Wavelet power spectrum along the red line in **(a)**, **(e)**

wavelet power spectrum along the red line in **(b)**, and **(f)** wavelet power spectrum along the red line in

**(c).**

## 3.2 Propagation of typhoon-induced CGWs in the mesosphere

As the typhoon moved along the coast of China, CGWs were identified at ten

stations in the OH network. Animation 1 shows that CGWs were observed by the OH

airglow network during 20:00–04:00 LT (the detailed data can be downloaded from the
Supplementary Material). As the weather conditions in North China during the study
period were better than those in South China, we identified clearer wave structures at the
northern stations   than at the southern stations. Nevertheless, circular wave structures
were visible for brief clear weather intervals at the Zhangzhou, Qujing, and Chongzuo
stations. The CGWs in the mesopause region extended to 2500 km, thereby nearly
covering the effective FOV of the OH airglow network.

As long as the CGWs do not encounter the critical layer or break, the CGWs

generated in the lower atmosphere   can propagate to the OH airglow layer. Through the
propagation group velocity, we can determine the propagation time to the OH layer. A
single dominant horizontal wavelength is seen at   each   altitude of 20 km, 40 km, and
60 km in the ERA-5 reanalysis. In contrast, the horizontal scales of the CGW obtained by
the OH airglow network were diverse, ranging from approximately 30 km to 300 km.
More importantly, we found some CGWs in the OH airglow layer, which were close to
the CGW wavelengths at 20 km, 40 km, and 60 km altitudes. To verify whether the same
wave was propagated from the reanalysis data layer to the OH layer, we used the group
velocity to estimate the time when the CGW at the altitudes of 20 km, 40 km, and 60 km
reached the OH airglow layer. The times required for the CGW in the three-layer
disturbance diagram in Fig. 4a, 4b, and 4c to reach the OH layer were approximately 21
minutes, 36 minutes, and 53 minutes. Therefore, the times when the CGWs visible in
ERA-5 at 60 km, 40 km, and 20 km would reach the OH airglow layer are approximately
23:21 LT, 23:36 LT, and 23:53 LT as shown in Fig. 5a, 5b, and 5c, respectively. The
wavelet analysis of Fig. 5f showed   that the horizontal wavelength of CGW in the OH
airglow layer (Fig. 5c) is approximately 156 km, the observed   period is approximately
23 min, and the horizontal speed is approximately 113 m/s, which is similar to the
dominant horizontal wavelength of the CGWs in the ERA-5 reanalysis at 20 km altitude.
Similarly, the horizontal wavelengths of CGW in the OH airglow layers (Fig. 5a and 5b)
were approximately 270 km and 295 km from the wavelet analysis of Fig. 5d and 5e,
which is similar to the dominant horizontal wavelength of the CGWs in the ERA-5
reanalysis at 60 km and 40 km altitudes. This suggests that the same CGW event can be
perfectly tracked over different altitudes and that the CGWs in the mesosphere
propagated upward from the stratosphere.

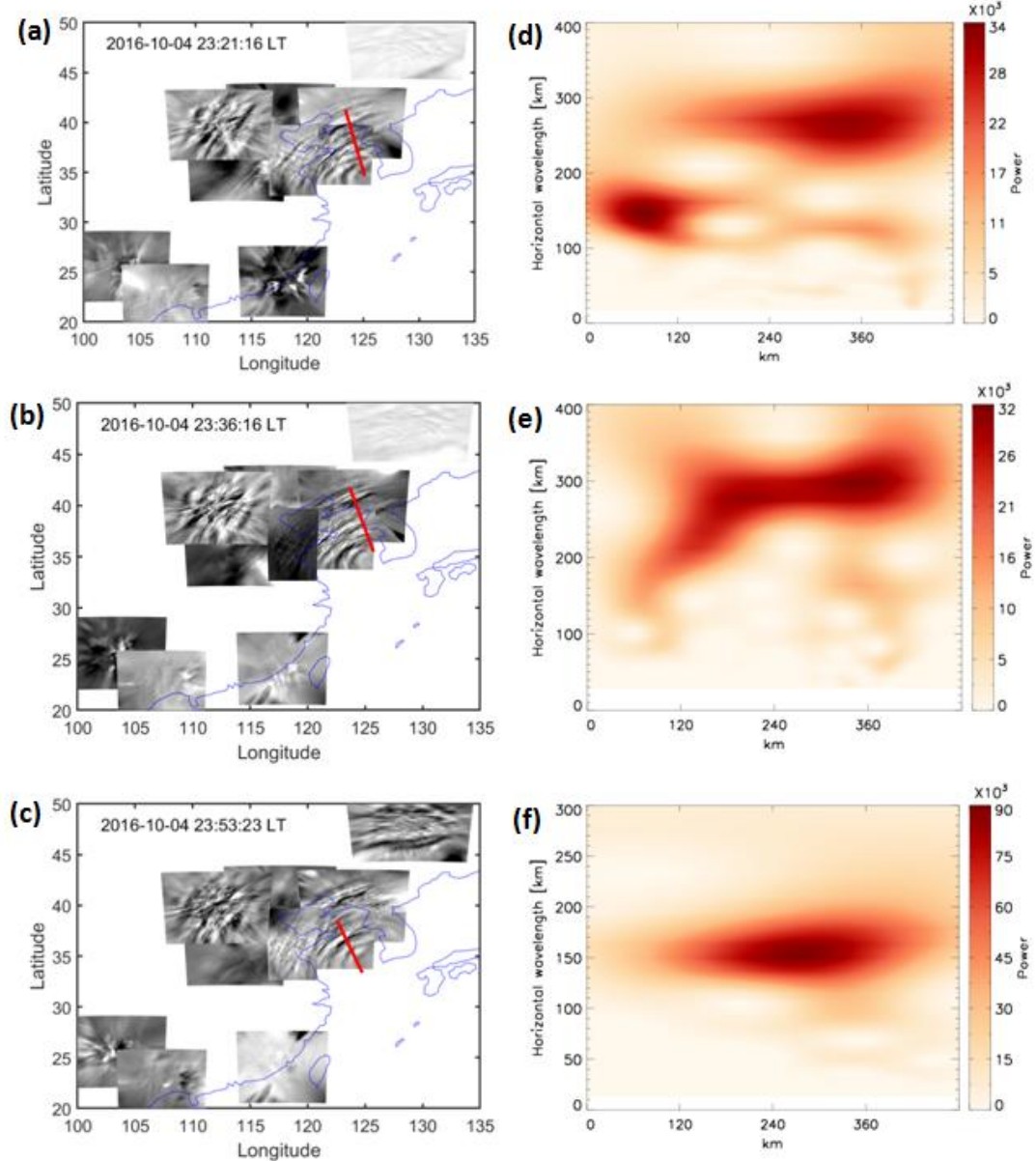

**Figure 5.** OH airglow emission perturbations induced by CGWs observed by the OH airglow imager network at **(a)** 23:21 LT, **(b)** 23:36 LT, and **(c)** 23:53 LT on 4 October 2016. **(d)** Wavelet power spectrum along the red line in **(a)**, **(e)** wavelet power spectrum along the red line in **(b)**, and **(f)** wavelet power spectrum along the red line in **(c).**

## 3.3 How typhoon-induced CGWs propagate to the thermosphere

Figure 6 shows the time sequence of the OI 630.0 nm airglow images from 00:57:05 LT to 01:12:22 LT on the night of 4 October 2016. Three curved phase fronts are clearly visible. The wave packet observed in the OI 630 nm airglow was quasi-monochromatic.

According to the wavelet analysis spectrum in Fig. 7, the horizontal wavelength was
approximately 120 km. The observed wave period and phase velocity were 10 min and
200 m/s, respectively. The horizontal wavelength was somewhat less than the
typhoon-induced concentric traveling ionosphere disturbances with a horizontal
wavelength from 160 to 200 km in the GNSS-TEC network as reported by Chou et al.
(2017). The CGW observed in the OI 630.0 nm airglow had much faster phase speed and
shorter period than that observed in the mesosphere, which indicate that its propagation
trajectory was relatively vertical. This means that they will not propagate as far
horizontally as the CGWs noted as dominant in the OH layer. Indeed, compared with the
long-distance extension of the CGWs in the mesosphere, the horizontal propagation
distance of the CGWs in the thermosphere was only 600 km from OI 630.0 nm network
observation. Vadas and Crowley (2010) showed that thermospheric GWs may be
secondary GWs generated by the breaking of primary GWs in the mesosphere and
thermosphere. We argue that the thermospheric CGW observed by the OI 630.0 nm
airglow imager was not directly generated by the typhoon, but a secondary GW. To test
this hypothesis, backward ray-tracing analysis was applied. In this way, we determined
the source of the CGW observed in the thermosphere.

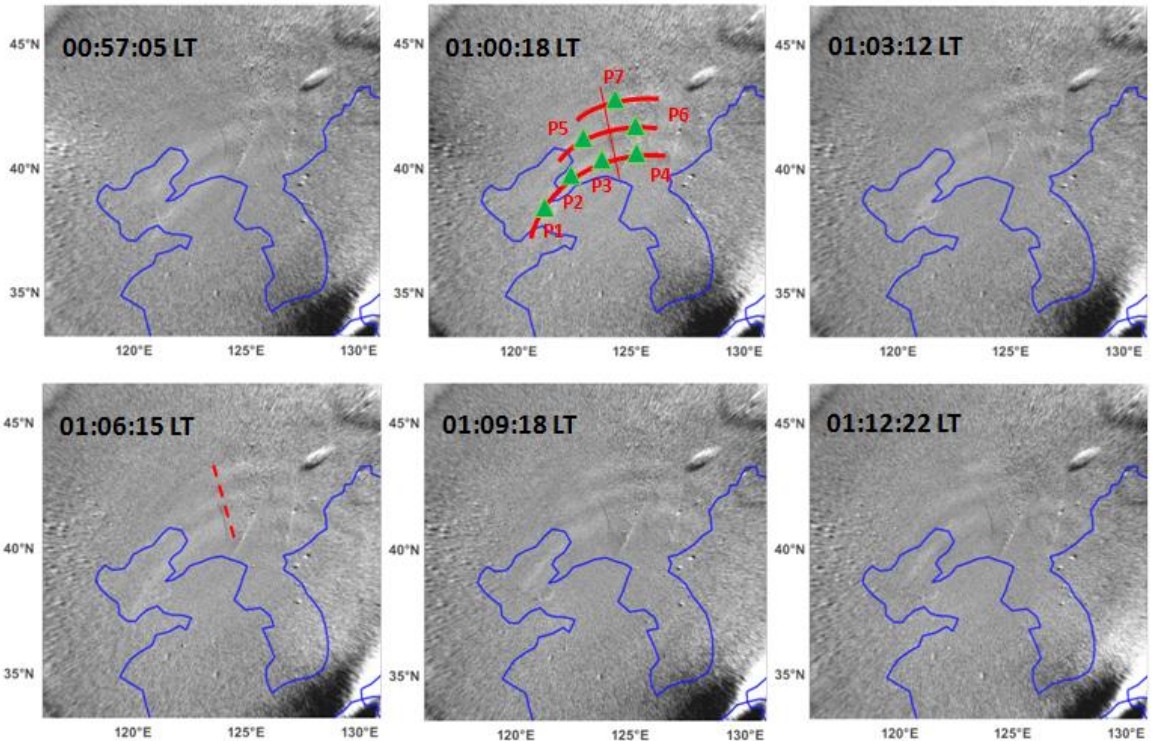


**Figure 6.** Time sequence of OI 630.0 nm airglow emission perturbation images observed by Donggng

station during 00:57:05 – 01:12:22 LT on the night of 4 October 2016. Green triangles (P1-P7) in the

red arcs are used as ray tracing sampling points. The blue line in each panel represents the coastline.

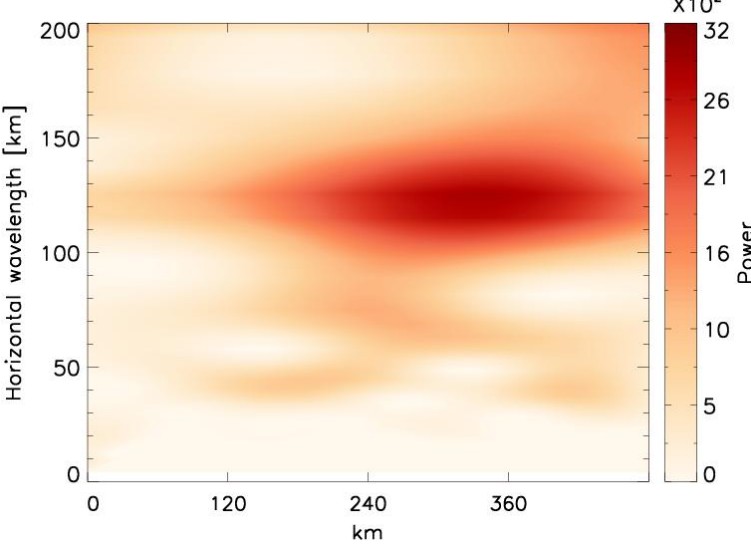


**Figure 7.** Wavelet power spectrum along the red line at 01:00:18 LT in Fig. 6.

We sampled seven points (green triangles) on a circular wavefront (red line in Fig. 6) at

01:00:18 LT as the starting point for backward ray tracing. The starting height of the

backward ray tracing was 250 km. The profile of the winds used in the ray tracing is
shown in Fig. 8a. The ray tracing trajectories of the seven sampling points are shown in
Fig. 8b. We used the following criterion to terminate the ray tracing: the square of the
vertical wavenumber should be negative. We started the ray-tracing at heights of 240 km,
250 km, and 260 km , and analyzed the results. The maximum uncertainty of horizontal
change of ray-tracing termination point caused by different starting heights was
approximately $\pm 0.36°$ in latitude and $\pm 0.17°$ in longitude (see Figure 8c). Subsequently,
seven backward traced trajectories took 37 minutes and terminated at an altitude of
approximately 95 km thereby indicating that a reflection layer was encountered.
According to linear theory, this suggests that the thermospheric CGW could not have
come from below 95 km. The thermospheric GW must have been generated at any
altitude between 95 km and the altitude of the OI 630.0 nm airglow. In other words, the
CGW observed in the thermosphere was excited after approximately 00:23 LT. Figure 9
presents the CGWs observed by the OH airglow network at 00:23:22 LT. We
superimposed the thermospheric CGWs along with the starting ray tracing points (green
triangles) reproduced from Fig. 6, and the backward ray tracing termination points (red
diamonds) on the OH airglow observation images. The dotted circle represents the
approximate fitting thermospheric CGW fronts. The center of the circle is marked by a
blue cross. Compared with the single-scale wave observed in the OI 630.0 nm layer,
multi-scale CGWs were visible from OH network observations. We found that the
termination points of ray tracing almost fell above the mesopause region. This suggests
that the CGW observed in the thermosphere did not directly originate from the typhoon
but may have emerged due to the dissipation and/or nonlinear processes of
typhoon-induced CGW in the mesopause region. However, the backward tracing terminal
positions (red diamonds in Fig. 9) did not coincide with the fitting circle center position
(blue cross in Fig. 9). Nevertheless, according to numerical simulation work by Vadas et
al. (2009), large winds can shift the apparent center of concentric rings from the location
of the convective plume. Indeed, we found strong southward winds from100 km to 140
km (with a peak value of 50 m/s at 150 km altitude) and from 160 km to 220 km (with a
peak value of 25 m/s at 175 km altitude) altitudes (right panel of Figure 8a). So the center
of the thermospheric CGW can be shifted southward from the location of the
thermospheric CGW sources in the mesopause region. For the zonal wind, the westward
wind dominated from the upper mesosphere to the thermosphere (left panel of Figure 8a).
Similarly, the thermospheric CGW center position shifted westward. Therefore, the
assumed center (blue cross) of the partial concentric ring GWs (blue arcs) actually shifted
to the southwest from the real source location, which may explain why the ray-tracing
result for the assumed GW source did not match the fitting center of the partial concentric
ring thermospheric GWs. Another possible mechanism is that the wave phase speeds are
accelerated by accelerating background winds. As mentioned above, the ground-based
frequency $\omega_r$ remains constant along a ray's path assuming the background wind field is
independent of time (Lighthill, 1978). However, transient effect (time derivatives of the
background wind components giving rise to time derivative of the frequency for a
particular ray) may cause the phase speeds to be accelerated, which may lead to the
ray-tracing results did not match the real locations. As the ray-tracing model used in this
study depended on the linear theory and did not consider the wave-wave and wave-mean
flow interactions and tunneling, the ray tracing results were limited and should be taken
into consideration carefully.

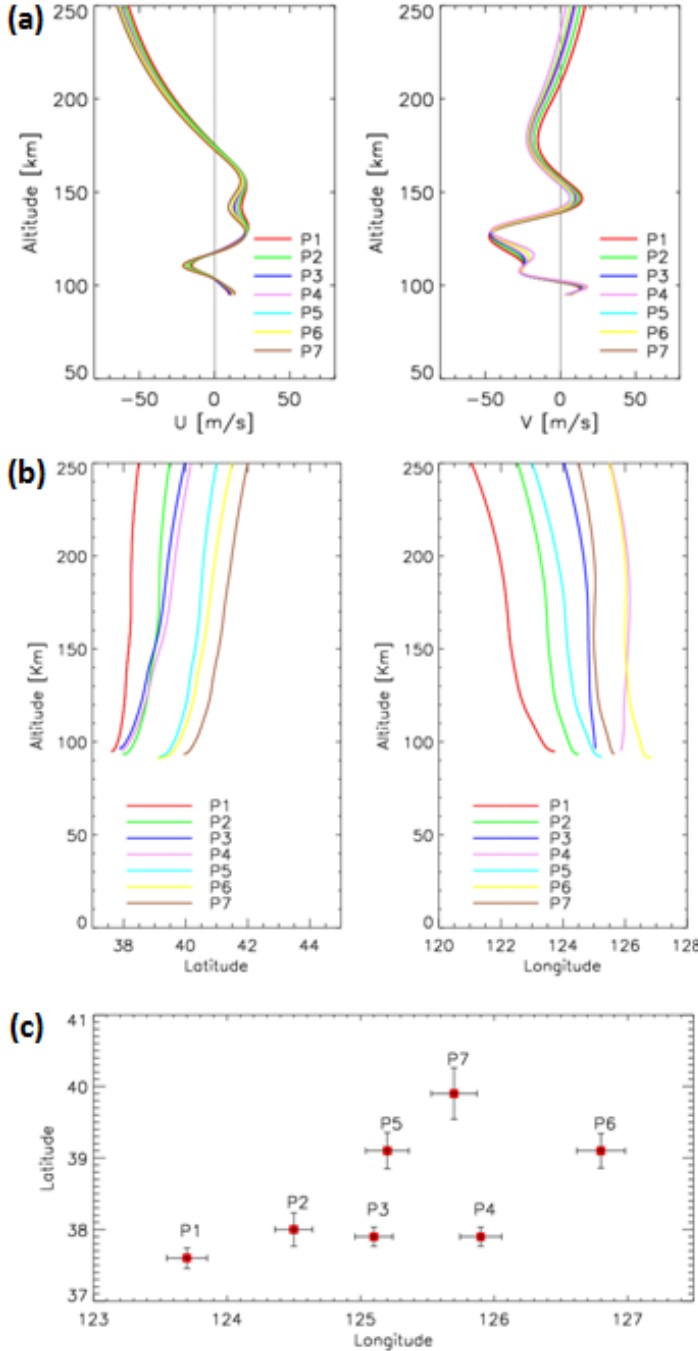


**Figure 8. (a)** Wind profiles along the seven ray-tracing paths. **(b)** Ray paths of the wave starting from

the seven sampling points in Fig.6. **(c)** Horizontal area distribution of the terminal positions of the

seven backward traced trajectories. Error bars give standard deviation for each point from the starting

altitude of 240 km, 250 km, and 260 km.

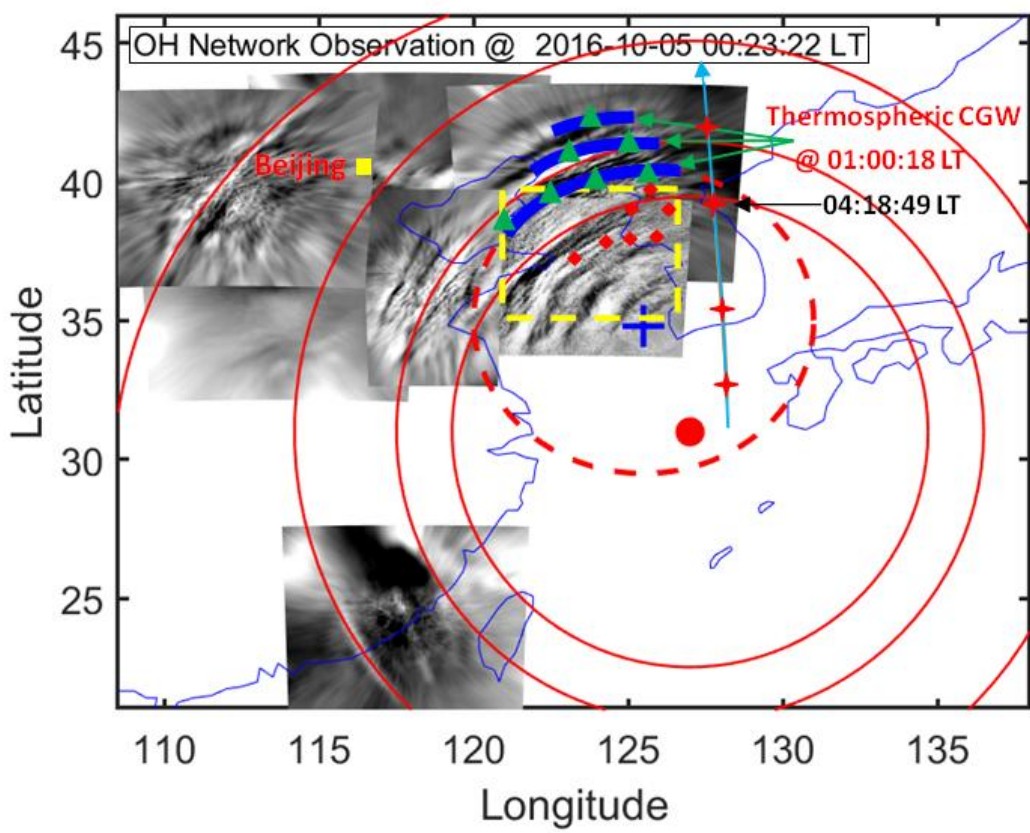


**Figure 9.** Double layer CGW superimposed graph: The blue arcs represent the thermospheric CGW

observed at 01:00:18 LT. The dotted circle represents the approximate fitting blue arcs. The blue cross

marks the center of the circle. The solid circles represent the approximate fitting CGWs observed by

the OH airglow network. The red dot marks the center of the circles. The green triangles and red

diamonds   represent the trace start and termination points, respectively. The red crosses represent the

sounding footprints of the TIMED/SABER measurements. The yellow box marks the location of the

meteor radar station.

## 4. Discussion

Figure 10 presents a time sequence of OH airglow images in the range marked by

the yellow dotted rectangle in Fig. 9. The images were retrieved from the Rongcheng

station from 00:01:30 to 00:44:30 LT on the night of 4 October 2016. At 00:01:30 LT,

three distinct curved wavefronts with horizontal wavelengths of approximately 96 km

were identified. Interestingly, wavefronts 2 and 3 collided and connected in the northeast,
indicating that wave-wave nonlinear interactions may have occurred.

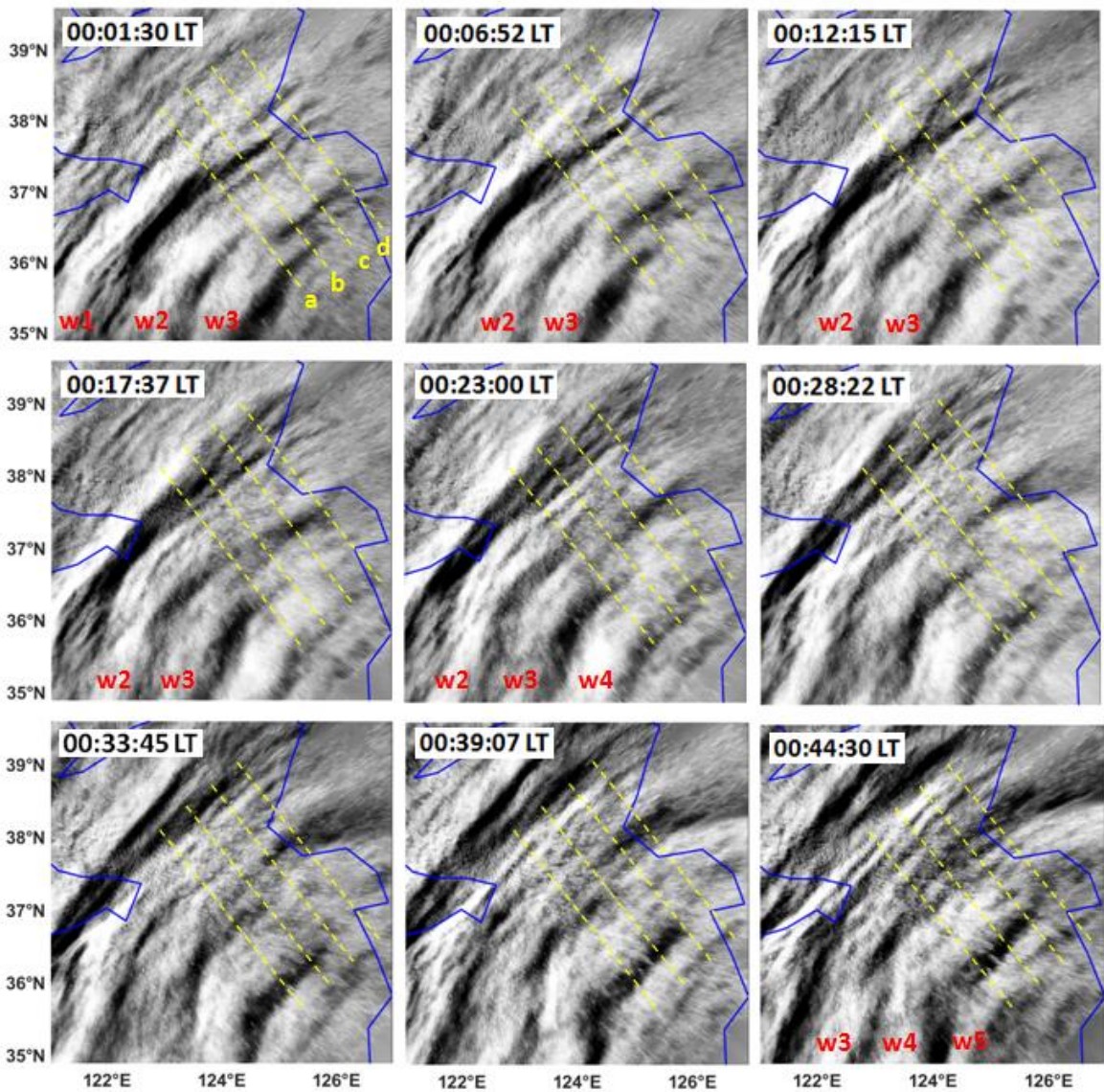


**Figure 10.** Time sequence of OH airglow emission perturbation images observed by Rongcheng
station during 01:01:30-00:44:30 LT on the night of 4 October 2016.w1-w5 denote the wavefronts of
the CGW. The blue line in each panel represents the coastline.

Figure 11 shows the time series of the OH image slices perpendicular to the

wavefronts (w1-w5). A dominant wavelength of approximately 150 km can be confirmed
at 00:00:25 LT. We found a significant attenuation of the amplitude from 00:00:25 LT to
00:17:37 LT. At 00:00:25 LT, while the relative average power was $2.3 \times 10^3$, and the
amplitude decreased gradually with time. At 00:17:37 LT, the average power decreased to
$0.15 \times 10^3$. We also identified the generation of approximately 110 km and 20-50 km
small-scale waves from the larger scales, which may be caused by wave-wave nonlinear
interactions and/or wave breaking. We overlayed the OI 630 nm airglow relative intensity
variation on the OH airglow variation and Figure 12 shows OH and OI 630 nm airglow
relative intensity variations. The OH plot was obtained at 00:29:27 LT and the OI 630 nm
plot at 01:06:15 LT. The time interval of 37 min was calculated by the above ray tracing
analysis. We obtained similar scale fluctuations were obtained in the two airglow layers.
The horizontal wavelength of the wave obtained by the OI 630 nm airglow layer was
approximately 118 km. The OH airglow layer has also obtained near-scale fluctuations
with wavelengths of approximately 109 km. These waves could be the same waves seen
in the thermosphere. Therefore, the CGW in the thermosphere may come from breaking
or nonlinear processes of that primary gravity waves.

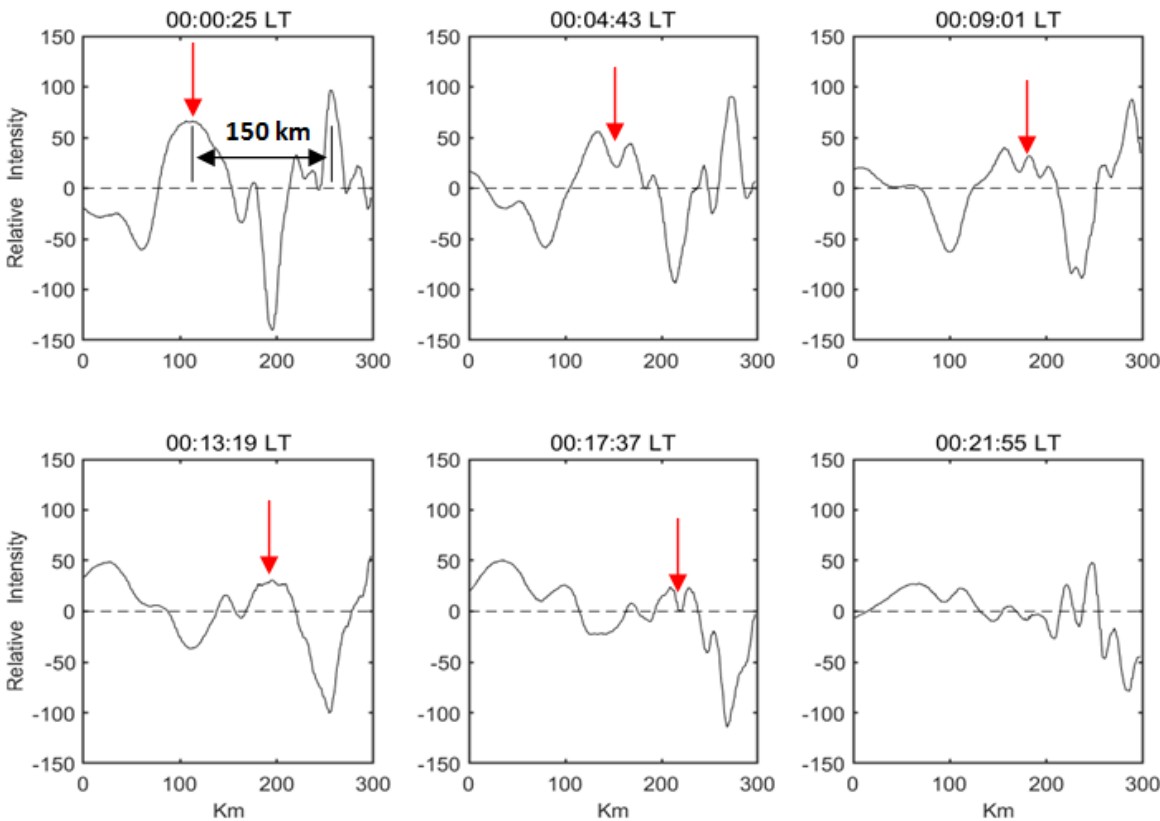

Figure 11. Time series of averaged OH image slices perpendicular to the wavefronts as marked by four yellow dotted lines (a, b, c, and d) in Fig.10. The wavefronts propagate from left to right. The red arrows mark the evolution of the wavefront peak.

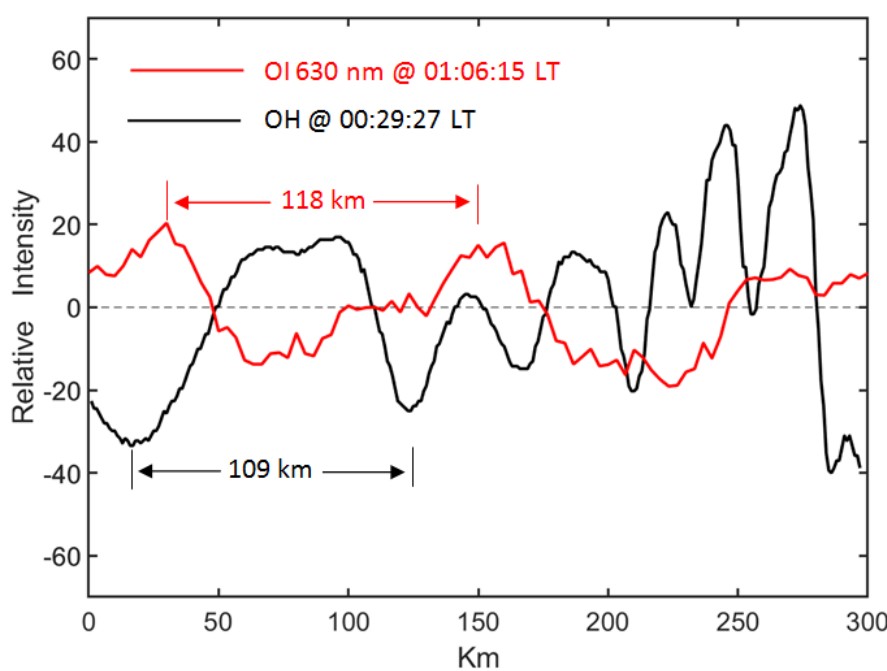

**Figure 12.** OH (black) and OI 630 nm (red) airglow relative intensity variations. The OH relative
intensity variation is obtained as in Fig. 11**.** The OI 630 nm relative intensity variation is from the red
dotted line in Fig.10 at 01:06:15 LT.

Note that wave   amplitude fluctuations can also result from the transient nature of

the wavepacket. The propagation state can be studied by using the dispersion relationship
with GW. However, the dissipation region of the CGW lacks the real-time background
temperature and wind field. In this context, the limb-viewing of Sounding of the
Atmosphere using Broadband Emission Radiometry (SABER) instrument on the
Thermosphere Ionosphere Mesosphere Energetics and Dynamics (TIMED) satellite can
be beneficial because it occurred near the wave-dissipation region; however, the time lag
was close to approximately 4 h. Background wind field data were obtained from an
ATRAD MDR6 all-sky VHF meteor radar at Beijing station. We further examined the
dispersion relationship of GW, thereby shedding some light on the possible propagation
state of dissipative waves. Figure 13 presents the vertical wave number $m^2$ profile derived
from the Beijing meteor radar wind and the temperature from the SABER/TIMED
measurement location at 04:18:49 LT, as marked in Fig. 9. The wave parameters used
were from the wavefronts (w1-w5) in Fig.10. The average horizontal wavelength was
approximately 96 km and the average observed phase velocity is approximately 90 m/s.
We identified a clear duct (from 87 km to 94 km) near the peak of the OH airglow layer.
Note that the duct can control the horizontal propagation of CGW. This implies that the
CGW may indeed be dissipated. In contrast, the upper boundary of the duct coincided
with the height of the ray-tracing termination area mentioned above. During wave
dissipation, momentum deposition occurs in the background atmosphere and can produce
bodyforces that stimulate secondary GWs (Fritts et al., 2006; Chun and Kim, 2008; Smith
et al., 2013; Vadas et al., 2018; Heale et al., 2020). In addition, secondary waves can be
generated by momentum transferred nonlinearly from the primary wave mode to
harmonics or subharmonics (Snively, 2017). Local momentum flux divergence associated
with wave breaking, vortex generation, and wave interactions can also generate
secondary GWs (Fritts et al., 2006).

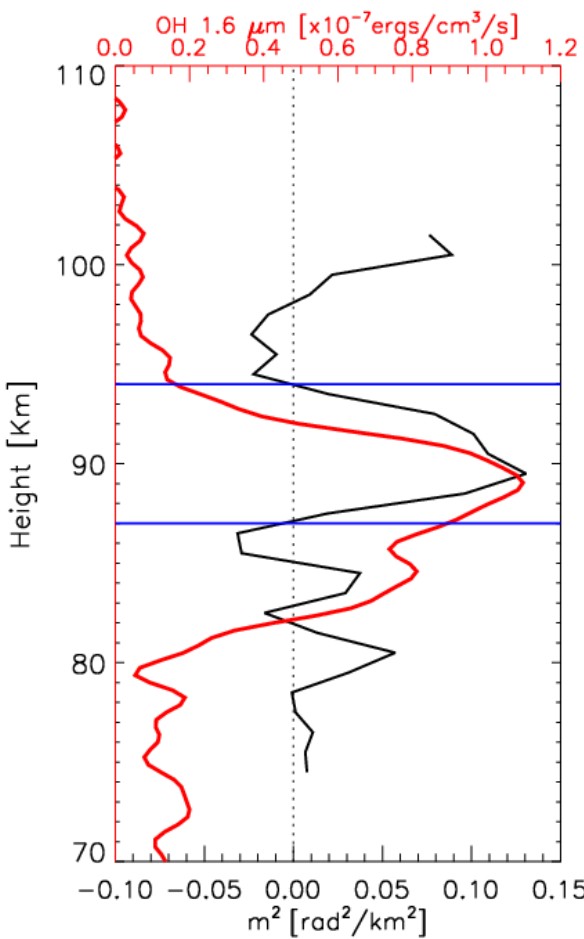


**Figure 13.** Vertical wave number m$^2$ profile (black) derived from the temperature from
TIMED/SABER measurement location at 04:18:49 LT and the meteor radar wind from Beijing station
marked in Fig. 9. The red line represents the OH1.6 μm emission intensity obtained by the
TIMED/SABER. The horizontal blue lines represent the top and bottom boundaries of the duct region.

## 5. Summary

In this study, a DLAN was used to capture CGWs over China that were excited by the Super Typhoon Chaba (2016). As Super Typhoon Chaba (2016) moved northward along the coast of the Chinese Mainland and developed to a mature stage, remarkable multi-layer CGW features produced by the Typhoon from near the ground to a height of 250 km were observed by ERA-5 reanalysis and airglow network. We applied the MTSAT-1R observations, ERA-5 reanalysis data, and backward ray tracing to quantitatively describe the physical mechanism of typhoon-generated CGWs propagating throughout the stratosphere, mesosphere, and thermosphere.

The temperature disturbance was approximately $\pm$1.5-2 K at 20 km and$\pm$3-4K at 40 km. However, the temperature disturbance ($\pm$1.3 K) at 60 km altitude did not increase with further increase in altitude, which may be caused by the sponge layer effect. Using reanalysis of multi-layer temperature disturbance, group velocity of gravity wave and wavelet analysis, we demonstrated that the CGWs in the mesopause region were excited directly by the typhoon.

Due to the observational limitations, a backward ray-tracing theory was used to connect GWs in the upper mesosphere to GWs in the thermosphere at about 250 km. We found that the termination points of ray tracing of the thermospheric CGW almost fell above the mesopause region. Backward ray-tracing analysis and the CGWs evolution process observed by the OH network suggested that the CGW observed in the thermosphere did not directly originate from the typhoon but may have emerged due to dissipation and/or nonlinear processes of typhoon-induced CGWs in the mesopause

region. Airglow network observations combined with numerical simulation to study the
generation of secondary wave in detail will be carried out in the future.

*Data availability*
The Double Layer Airglow Network data are available at http://159.226.22.74/. The
ERA-5 reanalysis data are downloaded from the Copernicus Climate Change Service
Climate Data Store through https://www.ecmwf.int/en/forecasts/datasets/
reanalysis-datasets/era5. The typhoon information are provided at
http://agora.ex.nii.ac.jp/digital-typhoon/. MTSAT-1R data is accessed from
http://webgms.iis.u-tokyo.ac.jp/.

*Video supplement*
A video of detailed evolutions of CGWs excited by the Typhoon observed by OH airglow
observation network is provided (https://doi.org/10.5446/55348).

*Author contributions*
J. X conceived the idea of the manuscript. Q. L. carried out the data analysis,
interpretation and manuscript preparation. H. L. L., X. L and W. Y. contributed to the data
interpretation and manuscript preparation. All authors discussed the results and
commented on the manuscript.

*Competing interests*

The authors declare no competing interests.

## *Acknowledgements*

This work was supported by the National Science Foundation of China (41974179 and 41831073), the Strategic Priority Research Program of Chinese Academy of Sciences (XDA17010301), the Informatization Plan of Chinese Academy of Sciences (CAS-WX2021PY-0101), and the Project of Stable Support for Youth Team in Basic Research Field, CAS (YSBR-018). The work was also supported by the Specialized Research Fund for State Key Laboratories. We acknowledge the use of data from the Chinese Meridian Project.

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
