# Peer review of "How do gravity waves triggered by a typhoon propagate from the"

_Atmospheric Chemistry and Physics, 2021_

## Referee Comment (RC2)

Thank you for the responses. I appreciate you taking the time to improve the manuscript based upon my comments. I still have a few comments on your corrections that I think may improve the manuscript further. I have shown your responses in red and commented in blue.

1.) Firstly, I believe the English in the title is not quite correct. If you can, I'd suggest changing it to: How do gravity waves triggered by a typhoon propagate from the troposphere to the upper atmosphere?

2.) "A single dominant horizontal wavelength of CGW at the altitude of 20 km, 40 km, and 60 km obtained by the ERA-5 reanalysis due to the limitation of resolution. In contrast the horizontal scales of CGW obtained by OH airglow network are diverse, ranging from approximately 30 km to 300 km."

I'd suggest re-wording as:  A single dominant horizontal wavelength is seen at the altitudes of 20 km, 40 km, and 60 km in the ERA-5 reanalysis due to the limited resolution. In contrast, the horizontal scales of the CGW obtained by OH airglow network are diverse, ranging from approximately 30 km to 300 km as the imager has much higher spatial resolution."

3.) Response:
We appreciate your careful review, which is very beneficial for improving our paper. Yes, you are right. There exist spectral powers between 100-150 km horizontal wavelength. But compared with the main spectral powers, they are very weak. The blue lines in Figure 9 in the revised manuscript also appear to match up with dark phases in the intensity of the OH image. However, the difference can be seen because the blue lines do not completely overlap with the fitted red circles.

I am still a little confused here. Both Figures 5d and f show strong power at ~150 km which to me suggests either possible harmonic generation of the 300km mode by nonlinear processes or these modes are simply part of the spectra. These waves could be the same waves seen in the Thermosphere, which also seem to have power around ~150km from the wavelet analysis you sent in the review response. In addition, figure 9 (assuming the x-axis is distance in km) also suggests a dominant wavelength of around ~150km (with smaller scales of ~25 km present which are likely wave breaking structures (e.g. in Heale et al. (2020) figures 5,7,8, 10 ) Can you overlay the OI 630 nm data on these plots to compare scales? My thinking is that primary gravity waves are breaking or nonlinearly generating these smaller scale secondary waves which are those seen in the thermosphere.

4. ) Response:
"In addition, the second waves can be generated by momentum transferred nonlinearly from the primary wave mode to harmonics or sub harmonics (Snively, 2017). Local momentum flux divergence associated with wave breaking, vortex generation, and wave interactions can also generate secondary GWs (Fritts et al., 2006)."

Change second waves to secondary wave

"Our analysis demonstrated that the CGWs in the mesopause region were directly by the typhoon, but the CGW observed in the thermosphere may be excited by the CGW dissipation and/or nonlinear processes in the mesosphere, rather than being directly excited by the typhoon and propagated to the thermosphere. Overall, the complete propagation process of the CGWs was studied and demonstrated. Specifically, it was shown how CGWs were generated by typhoon in the troposphere, passed through the stratosphere, reached the mesosphere. The obvious nonlinear wave-wave interaction and the dissipation process of CGWs are observed in the mesopause region. Therefore, momentum deposition due to wave dissipation and/or local momentum flux divergence associated with wave interactions generated secondary GWs,and then propagated to the thermosphere. "

I'd suggest re-writing as: Our analysis demonstrated that the CGWs in the mesopause region were excited directly by the typhoon, but the CGW observed in the thermosphere may be secondary wave excited by the primary CGW dissipation, breaking and/or nonlinear processes in the mesosphere, rather than being directly excited by the typhoon.

---

## Author Response (AR1)

**Referee #1-(Report 1)**

This paper uses multilayer observations (troposphere, mesosphere, and thermosphere) along with reanalysis data to characterize deep propagation of gravity waves generated above Super Typoon Chaba. The authors use wavelet analysis to examine the scales across the different layers and suggests that waves seen in the thermosphere could be secondary waves generated by wave dissipation in the mesosphere. This is determined by ray-tracing analysis and by examining the decay of wave amplitude in the OH layer. This paper is of interest to the community and can be published with just some relatively minor revisions. However, it could benefit significantly from some clarification and additions to the sections regarding ray-tracing, the link between the OH and 630 nm waves, and the explanations of wave dissipation and secondary wave generation processes. Comments are given below:

Thank you very much for your good comments concerning our manuscript entitled "How are the gravity waves triggered by typhoon propagate from the troposphere to upper atmosphere?". Those comments are all valuable and very helpful for revising and improving our paper, as well as the important guiding significance to our researches. We also thank you very much for your help in modifying the expression of English sentences. We have studied comments carefully and have made corrections which we hope meet with approval.

The detailed point-by-point responses are given below.

Line 23: Suggest removing 'the' from capture the concentric waves.

Response:
Thank you very much for your suggestion.
"the" is removed from "...... capture the concentric waves" in the revised manuscript.
(Please see line 23 in the manuscript with track)

Line 28: replace ray-tracing revealed that' with 'ray-tracing suggests that'

Response:
Thank you very much for your suggestion.
"revealed" is replaced by "suggests " in the revised manuscript.
(Please see line 29)

Line 30: What is meant by 'resembling the relay in the context'??

Response:
Thank you very much for your comment.
We re-described this sentence as:"However, like the relay, the backward ray tracing

analysis suggests that CGWs in the thermosphere originated from the secondary waves generated by the dissipation of the CGW and/or nonlinear processes in the mesopause region." in the revised manuscript.
(Please see line 28-30)

Line 45: Are a unique type of GWs. Note: concentric waves have also been generated from primary wave breaking, volcanoes, explosions, rockets (e.g. see works by Vadas and Becker ; Lund et al. (2020), Kogure et al (2020)).

Response:
Thank you very much for your valuable suggestions.
The following description is added to the revised manuscript.

"CGWs can also be generated by primary wave breaking (Vadas and Becker, 2019;Lund et al., 2020; Kogure et al., 2020) volcanoes (Duncombe,2022), explosions (Pierceet al.,1971), and rockets (Liuet al., 2020)."
(Please see line 41-44)

Line 45: 'convective activity'

Response:
"activities" is replaced by "activity" in the revised manuscript.
(Please see line 41)

Line 65: Suggest re-writing as: This paper presents a case study examining GCWs excited by Super Typhoon Chaba (2016).

Response:
Thank you very much for your good suggestions.
"This study examined the CGWs excited by Super Typhoon Chaba (2016) as a study case" is replaced by "This paper presents a case study examining CGWs excited by Super Typhoon Chaba (2016)"in the revised manuscript.
(Please see line 65-66)

Line 81: Just "mainland China"

Response:
"...the mainland China" is changed to "...mainland China"in the revised manuscript.
(Please see line 82)

Line 89: "With a central wavelength"

Response:
"...with the centre wavelength... " is replaced by "...with a central wavelength..."in

the revised manuscript.
(Please see line 90)

Line 116: add a space between October and 2016

Response:
a space is added between "October" and "2016"in the revised manuscript.
(Please see line 117)

Lines 142-143: resolution is used twice in this sentence.

Response:
 We appreciate your careful review.
"The horizontal resolution of the reanalysis temperature and wind data with a pre-interpolated resolution of 0.25° × 0.25° was used in this study." is changed to"
The horizontal reanalysis temperature and wind data with a pre-interpolated resolution of 0.25° × 0.25° was used in this study." in the revised manuscript.
(Please see line 141-143)

Line 147: Suggest "We use a ray tracing method to estimate the source location of the thermospheric secondary CGWs.

Response:
Thank you very much for your help.
"We use a ray tracing method to track the excitation source of the thermosphere secondary CGWs." is changed to " We use a ray tracing method to estimate the source location of the thermospheric secondary CGWs." in the revised manuscript.
(Please see line 147-148)

Line 176: add space to ERA-5reanalysis data.

Response:
a space is added between "ERA-5" and "reanalysis data" in the revised manuscript.
(Please see line 177-178)

Line 177: Why different times?

Response:
Thank you very much for your comment.
We set the time of the three layer temperature perturbations to 23:00LT.
(Please see Figure 4)

Figure 4: Axis labels.

Response:
Thank you very much for your careful review.
Axis labels are added to Figure 4in the revised manuscript.

[Figure]

**Figure 4.** Temperature perturbations at **(a)** ~60 km, **(b)** ~40 km, and **(c)** ~20 km at 23:00LT on 4 October 2016 derived from ERA-5 reanalysis.**(d)** The wavelet power spectrum along the red line in **(a), (e)** the wavelet power spectrum along the red line in **(b),** and **(f)** the wavelet power spectrum along the red line in **(c).**

Line 184: replaced "embraced" with "were present over a large area"

Response:
"embraced " is replaced by " were present over "in the revised manuscript.
(Please see line 186)

Line 208: How similar are the reanalysis datasets and the OH data? This seems a strange scientific decision. How do you justify this?

Response:
Thank you very much for your criticism.
What we want to express is the propagation of phase plane of CGWs from ERA-5 reanalysis datasets to the OH airglow layer. As long as the CGW does not encounter the critical layer or break, it can propagate to the OH airglow layer. Through the propagation group velocity, we can determine the propagation time to the OH layer.
(Please see line 211-214)

Line 208: also, I'm not sure single wavelength is correct, more a single wave packet, or a single dominant wavelength.

Response:
Thank you very much for your valuable suggestions. It should be more appropriate to describe wave with wave packet. "dominant" is added between "single" and "horizontal" in the revised manuscript.
(Please see line 214-215)

Line 211: Due to resolution of the reanalysis?

Response:
Thank you very much for your comment.
We have re-expressed this sentence as:"A single dominant horizontal wavelength is seen at the altitudes of 20 km, 40 km, and 60 km in the ERA-5 reanalysis due to the limited resolution. In contrast, the horizontal scales of the CGW obtained by OH airglow network are diverse, ranging from approximately 30 km to 300 km as the imager has much higher spatial resolution." in the revised manuscript.
(Please see line 214-218)

Line 213: You are talking about in the reanalysis dataset here?

Response:

Sorry, we didn't make it clear. We re-described it as:
"In order to verify whether the phase plane of the same wave is propagated from the reanalysis data layer to the OH layer," in the revised manuscript.
(Please see line 223-224)

Line 214: When the CGWs from ERA-5 at the altitudes….

Response:
"from ERA-5" is added between" CGWs" and "at the altitudes" in the revised manuscript.
(Please see line 229)

Line 214: Can you confirm that the waves in the ERA-5 at 20, 40, and 60km are the same wave? Perhaps using a meridional-vertical slice through the ERA-5 dataset to show continuity of wave phases with altitude.

Response:

Thank you for your suggestion. We want to express that the CGW scale on the specific reanalysis layer is consistent with that on the OH airglow layer due to the phase plane of the same wave propagating from the reanalysis data layer to the OH layer.

As your suggestion, a slice along wave propagation from the ERA-5 showing continuity of wave phases with altitude from 20 km to 60 km is given below:

Since the scale of CGW at the height of 20km is half that at the height of 40 km, it is not the same CGW spectrum.

[Figure]

Line 217: Usually waves generated from convective sources are part of a continuous spectrum of waves rather than discrete.

Response:
Yes, you are right.
Gravity waves generated by convective sources are often broad-spectrum. The

gravity wave observed at a certain height is only part of a continuous spectrum. We have re-described it as:

"Therefore, the time when the phase plane of CGWs from ERA-5 at the height of 60 km, 40 km, and 20 km reaches the OH airglow layer is approximately 23:28 LT, 23:39 LT, and 23:53 LT as shown in Fig. 5a, 5b, and 5c, respectively." in the revised manuscript.

(Please see line 228-231)

Lines 221, 224, and 225: Change 'observation period' to 'observed period'.

Response:

"observation " is changed to " observed "in the revised manuscript.

(Please see line 233)

Line 222, Line 226: change "horizontal wavelength of the atmosphere" to 'dominant horizontal wavelength of the CGWs in the ERA-5 reanalysis'

Response:

"horizontal wavelength of the atmosphere" is changed to "dominant horizontal wavelength of the CGWs in the ERA-5 reanalysis "in the revised manuscript.

(Please see line 234;239)

Line 224: I think a better sentence structure would be "The wave packet observed in the OI 630 nm airglow was quasi-monochromatic"

Response:

Thank you very much for your help.

"The wave scale observed at the OI 630.0 nm airglow was monochromatic." is replaced by "The wave packet observed in the OI 630 nm airglow was quasi-monochromatic ."in the revised manuscript.

(Please see line 257)

Figure 6: I am curious why you do not perform a wavelet analysis on the 630 nm data as you did with the OH and ERA-5 data? This may be beneficial for comparison

Response:

Thank you very much for your good suggestions.

We applied a wavelet analysis to the630 nm airglow data.

[Figure]

**Figure 7.** The wavelet power spectrum along the red line at 01:00:18 LT in Fig. 6.

Line 254: Maybe I am confused here? But the wavelet analysis in 5d, e, and f all show spectral power between 100-150km horizontal wavelength. The blue lines in Figure 7c also appear to match up with dark phases in the intensity of the OH image? The waves seen in the 630 nm will have relatively smaller amplitudes at OH heights and you wouldn't necessarily expect them to dominate there. The waves in the 630 nm data also have much faster phases speeds, shorter periods and thus a more vertical trajectory. This means that they will not travel as far horizontally as the waves noted as dominant in the OH layer so the radius of the concentric pattern would be expected to be smaller.

Response:
We appreciate your careful review, which is very beneficial for improving our paper.
Yes, you are right. There exist spectral powers between 100-150km horizontal wavelength. But compared with the main spectral powers, they are very weak. The blue lines in Figure 9in the revised manuscript also appear to match up with dark phases in the intensity of the OH image. However, the difference can be seen because the blue lines do not completely overlap with the fitted red circles.

[Figure]

**Figure 9.** Two layer superimposed graph: The blue arcs represent the thermospheric CGW observed at 01:00:18 LT. The dotted circle represents the approximate fitting blue arcs. The center of the circle is marked by a blue cross. The solid circles represent the approximate fitting CGWs observed by the OH airglow network. The center of the circles is marked by a red dot. The green triangles and diamonds represent the trace start and termination points, respectively. The red crosses represent the sounding footprints of the TIMED/SABER measurements. The yellow box marks the location of meteor radar station.

The following description is deleted from the manuscript.

"Nevertheless, the waves with a scale similar to that of the thermosphere GWs were not identified by the OH airglow network."

The following description is added to the new manuscript.

"The CGW observed in the OI 630.0 nm airglow having much faster phases speed and shorter period, which indicate that its propagation trajectory relatively vertical. This means that they will not propagate as far horizontally as the CGWs noted as dominant in the OH layer."
(Please see line 270-274)

Line 257: remove the word "moreover"

Response:
The word "moreover" is removed in the revised manuscript.
(Please see line 276)

Line 270: How sensitive is the result to the starting altitude and the phase of the 630 nm wave where the starting point for the reverse ray-trace is chosen?

Response:
Yes, you are right.
Uncertainty analysis is necessary for scientific research.
We added three more starting points(see Figure 6 at 01:00 18 LT) in different phases. The ray tracing results of three different heights of 240 km, 250 km and 260 km are analyzed.

We find that the termination points of ray tracing almost fall in the dissipative and/or nonlinear processes region or at the edge of the region. The maximum uncertainty of horizontal change of ray tracing termination point caused by different starting heights is approximately ± 0.36° in latitudinal and ± 0.17° in longitudinal (see Figure 8c).

[Figure]

**Figure 6.** A time sequence of OI 630.0 nm airglow images observed by Donggng station during 00:57:05-01:12:22 LT on the night of 4 October2016. Green triangles (P1-P7) in the red arcs are used as ray tracing sampling points. The blue line in each panel represents the coastline.

[Figure]

**Figure 8. (a)** The wind profiles along the four ray-tracing paths. **(b)** The ray paths of the wave starting from the seven sampling points in Fig.6. **(c)** Horizontal area distribution of the terminal positions of the seven backward traced trajectories. Error bars give standard deviation for each point from the starting altitude of 240km, 250km, and 260km.

Line 275: Just because the wave would have been reflected at 95 km there doesn't necessary mean it was generated there. It just suggests that the wave could not have comes from below this altitude according to linear theory. However, the wave could have been generated at any altitude between 95 km and the altitude of observation.

Response:
Thank you very much for your criticism.
We have re-described this issue in the new manuscript
"Subsequently, four backward traced trajectories took 37 minutes and terminated at the altitude of approximately 95 km thereby indicating that it met the reflection layer, which suggests that the thermospheric CGW could not have comes from below95 km according to linear theory. the thermospheric CGW could have been generated at any altitude between 95 km and the altitude of the OI 630.0 nm airglow. In other words, the CGW observed in the thermosphere was excited after approximately 00:23 LT."
(Please see line 298-304)

Line 315: I think the images in Figure 8 show clear signs of nonlinearity, instability, and smaller scale wave generation via wave-wave interaction and/or wave breaking.

Response:
Thank you for your constructive suggestions.
The following description is added to the revised manuscript.

"Interestingly, the wavefronts 2 and 3 collided and connected in the northeast, indicating that wave-wave nonlinear interactions may have occurred."

"At the same time, we also identified the generation of approximately 110 km and 20-50km small-scale waves from the larger scales, which may be caused by wave-wave nonlinear interactions and/or wave breaking."
(Please see line 343-344;356-358)

Line 316: I'd suggest rewriting as: "However, it is noted that wavepacket amplitude fluctuations can also result from the transient nature of the wavepacket."

Response:
"However, the observed CGW dissipation may be caused by the upward CGW passing through the airglow." is changed to "However, it is noted that wavepacket amplitude fluctuations can also result from the transient nature of the wavepacket." in the revised manuscript.
(Please see line 376-377)

Line 317: I'm not sure I understand the statement: "the observed CGW dissipation is real, unless it propagates horizontally"

Response:
We found the following description redundant, so the following description is deleted from the manuscript.

"Notably, the observed CGW dissipation is real, unless it propagates horizontally."

Line 328: Why does this imply that the CGW may be dissipated? I think the Figure 8 with its evidence of small-scale structure and nonlinearity is evidence enough. What wave parameters were used for the m2 analysis in Figure 10?

Response:
Because the observed amplitude change is not only caused by dissipation, but also by a wave packet passing through the layer as it propagates upwards. Of course, this is only a possibility. According to your suggestion, we also discussed the nonlinear interaction processes such as wave-wave interaction and/or wave breaking.

The wave parameters used for the m2 analysis in Figure 13 are from the wavefronts (w1-w5) in Figure 10. The average horizontal wavelength is approximately 96 km and the average observed phase velocity is approximately 90 m/s.
(Please see line 388-390)

Lines 330-332: While this statement is true, this is a huge simplification of the problem. Secondary waves can be generation via many mechanisms that are both non-dissipative and dissipative. Momentum does not need to be deposited in the mean flow to generate secondary waves but can be transferred nonlinearly from the primary wave mode to harmonics or subharmonics, the wave can also induce local mean flow accelerations just because of the transience of the wave packet which can lead to wave breaking. Local momentum flux divergence associated with wave breaking, vortex generation, and wave interactions can also generate secondary acoustic and gravity waves. For some references see: Franke and Robinson (1999), Fritts et al (2006), Zhou et al (2002), Chun and Kim (2008), Lund and Fritts (2012), Fritts et al (2015), Dong et al. (2020), Fritts et al (2020), Heale et al (2020; 2021), Bolini et al (2016), Vadas et al (2003), Vadas and Becker (2019), Scinocca and Ford 2000, Snively 2017. I'd recommend sections 2.2 and 4 from Fritts et al. (2006): Mean and variable forcing of the middle atmosphere by gravity waves.

Response:

Thank you for your serious criticism. We have made a more complete discussion.
The following description is added to the section 4 of the revised manuscript.

"In addition, the secondary wave can be generated by momentum transferred nonlinearly from the primary wave mode to harmonics or subharmonics (Snively, 2017). Local momentum flux divergence associated with wave breaking, vortex generation, and wave interactions can also generate secondary GWs (Fritts et al., 2006)."
(Please see line 396-400)

Relevant contents have also been modified in the Abstract:

"However, like the relay, the backward ray tracing analysis suggests that CGWs in the thermosphere originated from the secondary waves generated by the dissipation of the CGW and/or nonlinear processes in the mesopause region".

and in the Summary:

"Our analysis demonstrated that the CGWs in the mesopause region were excited directly by the typhoon, but the CGW observed in the thermosphere may be secondary wave excited by the primary CGW dissipation, breaking and/or nonlinear processes in the mesosphere, rather than being directly excited by the typhoon from backward ray tracing analysis and the CGWs evolution process observed by OH network."

**Referee #1-(Report 2)**

Thank you for the responses. I appreciate you taking the time to improve the manuscript based upon my comments. I still have a few comments on your corrections that I think may improve the manuscript further. I have shown your responses in red and commented in blue.

Thank you very much for your further comments concerning our manuscript. Those comments are all valuable and very helpful for revising and improving our manuscript, as well as the important guiding significance to our researches. We have studied comments carefully and have made corrections which we hope meet with approval.

The detailed point-by-point responses are given below.

1. Firstly, I believe the English in the title is not quite correct. If you can, I'd suggest changing it to: How do gravity waves triggered by a typhoon propagate from the troposphere to the upper atmosphere?

Response:
Thank you very much for your good suggestion.
"How are the gravity waves triggered by typhoon propagate from the troposphere to upper atmosphere?" is changed to "How do gravity waves triggered by a typhoon propagate from the troposphere to the upper atmosphere?" in the revised manuscript.

2. "A single dominant horizontal wavelength of CGW at the altitude of 20 km, 40 km, and 60 km obtained by the ERA-5 reanalysis due to the limitation of resolution. In contrast the horizontal scales of CGW obtained by OH airglow network are diverse, ranging from approximately 30 km to 300 km."

I'd suggest re-wording as: A single dominant horizontal wavelength is seen at the altitudes of 20 km, 40 km, and 60 km in the ERA-5 reanalysis due to the limited resolution. In contrast, the horizontal scales of the CGW obtained by OH airglow network are diverse, ranging from approximately 30 km to 300 km as the imager has much higher spatial resolution."

Response:
"A single dominant horizontal wavelength of CGW at the altitude of 20 km, 40 km, and 60 km obtained by the ERA-5 reanalysis due to the limitation of resolution. In contrast the horizontal scales of CGW obtained by OH airglow network are diverse, ranging from approximately 30 km to 300 km." is changed to "A single dominant horizontal wavelength is seen at the altitudes of 20 km, 40 km, and 60 km in the

ERA-5 reanalysis due to the limited resolution. In contrast, the horizontal scales of the CGW obtained by OH airglow network are diverse, ranging from approximately 30 km to 300 km as the imager has much higher spatial resolution.".
(Please see line 214-218)

3. We appreciate your careful review, which is very beneficial for improving our paper. Yes, you are right. There exist spectral powers between 100-150 km horizontal wavelength. But compared with the main spectral powers, they are very weak. The blue lines in Figure 9 in the revised manuscript also appear to match up with dark phases in the intensity of the OH image. However, the difference can be seen because the blue lines do not completely overlap with the fitted red circles.

I am still a little confused here. Both Figures 5d and f show strong power at ~150 km which to me suggests either possible harmonic generation of the 300 km mode by nonlinear processes or these modes are simply part of the spectra. These waves could be the same waves seen in the Thermosphere, which also seem to have power around ~150 km from the wavelet analysis you sent in the review response. In addition, figure 9 (assuming the x-axis is distance in km) also suggests a dominant wavelength of around ~150 km (with smaller scales of ~25 km present which are likely wave breaking structures (e.g. in Heale et al. (2020) figures 5,7,8, 10 ) Can you overlay the OI 630 nm data on these plots to compare scales? My thinking is that primary gravity waves are breaking or nonlinearly generating these smaller scale secondary waves which are those seen in the thermosphere.

Response:

We are extremely grateful to you for pointing out this issue.
Yes, you are right. Figures 5d and f do show strong power at ~150 km. As your suggestion, we compared the wave scales in OH and OI 630 nm by overlay the OI 630 nm data on OH plot.

The following description is added to the revised manuscript.

"We also overlay the OI 630 nm airglow relative intensity variation on OH airglow variation. Figure 12 shows OH and OI 630 nm airglow relative intensity variations. The OH plot is obtained at 00:29:27 LT and the OI 630 nm plot is obtained at 01:06:15 LT. The time interval of 37 min is calculated by the above ray tracing analysis. We found that similar scale fluctuations were obtained in the two airglow layers. The horizontal wavelength of the wave obtained by OI 630 nm airglow layer is approximately 118 km. The OH airglow layer has also obtained near scale fluctuations with a wavelength of approximately 109 km. Therefore, the CGW seen in the thermosphere may suggest come from breaking or nonlinear processes of that primary gravity wave."
(Please see line 358-366)

[Figure]

**Figure 12.** OH (black) and OI 630 nm (red) airglow relative intensity variations. The OH relative intensity variation is obtained as Fig. 11. The OI 630 nm relative intensity variation is from red dotted line in Fig.10 at 01:06:15 LT.

4. "In addition, the second waves can be generated by momentum transferred nonlinearly from the primary wave mode to harmonics or sub harmonics (Snively, 2017). Local momentum flux divergence associated with wave breaking, vortex generation, and wave interactions can also generate secondary GWs (Fritts et al., 2006)."

Change second waves to secondary wave.

Response:
" second waves " is changed to "secondary wave".
(Please see line 396)

5. "Our analysis demonstrated that the CGWs in the mesopause region were directly by the typhoon, but the CGW observed in the thermosphere may be excited by the CGW dissipation and/or nonlinear processes in the mesosphere, rather than being directly excited by the typhoon and propagated to the thermosphere. Overall, the complete propagation process of the CGWs was studied and demonstrated. Specifically, it was shown how CGWs were generated by typhoon in the troposphere, passed through the stratosphere, reached the mesosphere. The obvious nonlinear

wave-wave interaction and the dissipation process of CGWs are observed in the mesopause region. Therefore, momentum deposition due to wave dissipation and/or local momentum flux divergence associated with wave interactions generated secondary GWs, and then propagated to the thermosphere. "

I'd suggest re-writing as: Our analysis demonstrated that the CGWs in the mesopause region were excited directly by the typhoon, but the CGW observed in the thermosphere may be secondary wave excited by the primary CGW dissipation, breaking and/or nonlinear processes in the mesosphere, rather than being directly excited by the typhoon.

Response:

This description has been re-described according to your suggestion.

(Please see line 415-419)

**Referee #2**

The subject of the manuscript is rather interesting and there is a lot of new information about gravity waves generation, propagation and dissipation. The powerful meteorogical source should generate the concentric gravity waves and the all-sky camera network together with reanalysis data very convincing demonstrate it. It will be very interesting to all in the Earth atmospheric research community to read the article.

Response:
Thank you very much for taking your time to review our manuscript. Thank you very much for your very positive and constructive comments.

---

## Author Response (AR2)

Dear Editor and Referee:

We would like to take this opportunity thank to you for taking time off your busy schedules to review the manuscript.

We have completed the comments of Referee #3. The comments of Referee #3 are based on the online discussion Version (acp-2021-952-manuscript-version3.pdf), rather than the revised Version (acp-2021-952-manuscript-version4.pdf) based on the comments of Referee #1 and Referee #2. The main comments concerned by Referee #3 have been considered in the revised Version (acp-2021-952-manuscript-version4.pdf). Nevertheless, we have reconsidered carefully all the comments raised by Referee #3.

Thank you for your consideration. We look forward to hearing from you.

Sincerely,

Jiyao Xu

**Referee #3**

General comment:

This paper presents results on GW propagation from the lower to the upper atmosphere during the 2016 typhoon Chaba. The applied methods include 1) reanalysis, 2) airglow emissions from the OH in the mesopause region and from the OI 630 nm emission (emission height ~250 km), and 3) ray tracing calculations of GWs. Due to the coverage of the mainland of China with corresponding instruments, 2D images of gravity wave (GW) signatures in the OH and OI emissions can be linked to GW signatures in the stratosphere as seen in ERA5-reanalysis. The major conclusions from this paper are that GW images in the mesopause region are found to be consistent with stratospheric GWs in reanalysis, and that GW signatures in the thermosphere can be explained by assuming that these waves were generated around or above the mesopause. In this respect, the present paper addresses a current hot topic in the atmospheric dynamics community, namely the mechanism of "multi-step vertical coupling" (Vadas and Becker, 2019), which means that GW effects at higher altitudes are often not directly due to primary GWs generated in the troposphere, but are due to higher-order GWs. The present paper manuscript, however, has a number of significant shortcomings that need to be addressed/solved before this study can be considered for publication in ACP. These shortcomings pertain to 1) the writing (the English and the arrangements of thoughts are often confusing, the citations are incomplete, and the summary section is insufficient), 2) the methods by which the major conclusions are derived, and 3) contradictions of the conclusions to previous studies and even to results within this present paper.

Thank you very much for your good comments concerning our manuscript. Those comments are all valuable and very helpful for revising and improving our paper, as well as the important guiding significance to our researches. We have studied comments carefully and have made corrections which we hope meet with approval.

The detailed point-by-point responses are given below.

Major comments:

**1 The English is often confusing. It is beyond the scope of this review to name all the places in the manuscript to which this comment applies and make suggestions. I strongly recommend that the paper should be carefully iterated by a native English speaker after a substantial revision is completed.**

The arrangement of thoughts is sometimes confusing as well. As an example, I would like mention that some introductory sentences (e.g., L171-173, L296-298, L236-238) seem to refer to results and conclusions already made. Then, one or two paragraphs later the reader has to learn that the authors simply anticipated some conclusion or summary statement related to results that had yet to be presented in the respective section.

The writing and the content of the summary section appears to be insufficient.

Response:

Thank you very much for your comments.

According to your review opinions, we have improved the English writing and arrangement of the whole manuscript by the Wiley Editing Services. Also, we give a more sufficient description in the summary part of the manuscript.

The language editing certificate is attached at the end.

**2 A main conclusion of this paper is that the stratospheric GWs shown in Fig. 4c (having a predominant horizontal wavelength of ~156 km) reach the mesopause region prior to the larger-scale GWs seen Fig. 4a (having a predominant horizontal wavelength of ~295 km). This conclusion is not consistent with Figs. 4b, c, which suggest that concentric GWs having larger horizontal scales are seen earlier at higher altitudes. Note that a body of studies by Vadas and colleagues exist about the propagation characteristics of concentric GWs (e.g., Vadas et al. (2012), Yue et al. (2009), Vadas and Azeem (2021)). Some of these studies are even cited in the current manuscript. According to these former studies, the concentric GWs from convective sources that have larger horizontal wavelengths propagate faster to higher altitudes and are less prone to dissipation. The reason is that the GWs from such sources with larger horizontal wavelengths also have larger vertical wavelengths and, therefore, larger vertical group velocities. The conclusion made on page 13 of the paper contradicts these former results (and Figs. 4b, c as well).**

Response:

Thank you very much for your comments. I'm very sorry that this issue has not been clearly expressed.
Because the time of reanalysis data of three layers is inconsistent, the reanalysis data of 20 km and 40 km altitude is 23:00 LT, and the reanalysis data of 60 km is 24:00 LT (Please check Figure 4 of the manuscript version you reviewed), so it seems like that the CGW in 60 km layer propagates slowly.
In order to more clearly show the propagation characteristics of gravity waves with different scales at different altitudes,we set the time of the three layer temperature perturbations to 23:00 LT in the revised manuscript.

(Please check Figure 4 and line 207-231 in the revised manuscript)

In this context the authors may notice that the temperature perturbations shown in Fig. 4 from reanalysis are extremely small compared to other estimates of typical stratospheric GWs. For example, Becker et al. (2022) showed that typical temperature perturbation amplitudes simulated by a high-resolution GCM in the wintertime lower stratosphere are +- 1-2 K, and about +- 5 K in the stratopause region. For a major typhoon we would expect even larger amplitudes. Figure 4, on the other hand, shows GW perturbation amplitudes from reanalysis that are too weak by at least a factor of 100 in the stratopause region! It is well known that reanalyses generally underestimate the stratospheric GWs by a significant amount. Furthermore, a height of 60 km (Fig. 4a) appears to be well within the sponge layer of the reanalysis model (the GWs amplitudes DECREASE with height in Figs. 4a,b,c by a factor of 5 from the lower to the upper stratosphere). The authors did not take into account or discuss these deficiencies. Indeed, the realism of the concentric GW structures shown in Fig. 4 seems very questionable. Hence, the concentric GWs seen in OH airglow (Fig. 5) are likely not the same GWs as those shown in Fig.4.

Response:
Thank you very much for your comments. Yes, you are right. The real temperature disturbances shown in Fig. 4 is wrong.
Because we only want to show the gravity wave more clearly, we ignore the display of the real temperature disturbance. When we remove the background, the sliding window is too small, so the background is not completely removed. We recalculated the temperature disturbance. Temperature perturbations were calculated by subtracting the background with a 7 ×7 grid point running mean at 20 km and 17 ×17 grid point running mean at 40 km and 60 km. We found that the temperature disturbance was about $\pm$1.5－2 K at 20 km and $\pm$3－4K at 40 km. Using the ECMWF reanalysis data, Kim et al.(2009) reported a similar temperature disturbance($\pm$4K) at 40 km altitude. Becker et al. (2022) showed that typical temperature perturbation amplitudes simulated by a High Altitude Mechanistic general Circulation Model were ±1-2K in the wintertime lower stratosphere and ±5 K in the stratopause region. However, the temperature disturbance at 60 km altitude was only $\pm$1.3 K and did not increase with increasing altitude, which may be caused by this altitude being well within the sponge layer of the reanalysis model.

(Please check Figure 4 and line 168-180 in the revised manuscript)

**3 The connection from the upper mesosphere and GWs to the thermosphere is made via backward ray tracing of GWs seen in the OI emissions. Figure 7c indicates the corresponding concentric ring structures in OI. According to the aforementioned studies of concentric GWs, the center of the red rings in Fig. 7c should correspond to the geographical location of the assumed GW source. The authors argue that this source is in the mesopause region where the primary waves from the typhoon presumably dissipate. However, the backward rays (red lines in that Fig. 7c) end very far away from the center of the rings. In other words, the ray tracing result for the**

assumed GW source and the assumed center of the center of the partial concentric ring GWs in Fig. 7c do not match at all. This mismatch is not even mentioned in the present manuscript.

Response:

I'm very sorry that this issue has not been clearly expressed. The fitting center of the thermospheric CGW (blue arcs) is a blue cross rather than a red dot. The red dot is the fitting center of the CGW (solid circles) in the OH layer. Please See Figure 9 below.

However, the backward tracing terminal positions (red diamonds in Fig. 9) did not coincide with the fitting circle center position (blue cross in Fig. 9). Nevertheless, according to numerical simulation work by Vadas et al. (2009), large winds can shift the apparent center of concentric rings from the location of the convective plume. Indeed, we found strong southward winds from100 km to 140 km (with a peak value of 50 m/s at 150 km altitude) and from 160 km to 220 km (with a peak value of 25 m/s at 175 km altitude) altitudes (right panel of Figure 8a). So the center of the thermospheric CGW can be shifted southward from the location of the thermospheric CGW sources in the mesopause region. For the zonal wind, the westward wind dominated from the upper mesosphere to the thermosphere (left panel of Figure 8a). Similarly, the thermospheric CGW center position shifted westward. Therefore, the assumed center (blue cross) of the partial concentric ring GWs (blue arcs) actually shifted to the southwest from the real source location , which can explain why the ray-tracing result for the assumed GW source did not match the fitting center of the partial concentric ring thermospheric GWs.

The above description is added to the revised manuscript.

(Please check Figure 8a and 9 and line 278-305 in the revised manuscript)

[Figure]

**Figure 8. (a)** The wind profiles along the seven ray-tracing paths.

[Figure]

**Figure 9.** Double layer CGW superimposed graph: The blue arcs represent the thermospheric CGW observed at 01:00:18 LT. The dotted circle represents the approximate fitting blue arcs. The blue cross marks the center of the circle. The solid circles represent the approximate fitting CGWs observed by the OH airglow network. The red dot marks the center of the circles. The green triangles and red diamonds represent the trace start and termination points , respectively. The red crosses represent the sounding footprints of the TIMED/SABER measurements. The yellow box marks the location of the meteor radar station.

Other comments:

The citation is not sufficient regarding the original papers of higher-order GW generation and their effects in the thermosphere/ionosphere. Indeed, the mathematical theory for higher-order GWs was derived in Vadas et al (2003), and a summary of that theory and its implications was given in Vadas et al. (2018). Furthermore, global simulations of concentric higher-order GWs in the thermosphere were first discussed in Vadas and Becker (2019).

Figs. 5a,b,c, 6, 7c, 8: The figure captions do not mention the physical quantities that are shown. Also the color bars with corresponding units are missing.

Response:

The references below are added to the list of references of the revised manuscript and discussed appropriately.

The physical quantities are all shown in the figure caption descriptions. The color bars with corresponding units are all added.

References:

Vadas et al (2003): Mechanisms for the generation of secondary waves in wave breaking regions, J. Atmos. Sci., 60, 194-214.

Vadas, S. L., Yue, J., She, C. Y., Stamus, P., and Liu, A. Z.: A model study of the effects of winds on concentric rings of gravity waves from a convective plume near Fort Collins on 11 May 2004, J. Geophys. Res., 114, 2009.

Vadas et al (2012): Mesospheric concentric gravity waves generated by multiple convective storms over the North American Great Plain. JGR, 117, doi:10.1029/2011JD017025.

Vadas et al. (2018): The excitation of secondary gravity waves from local body forces: Theory and observation, J. Geophys. Res. Atmos., doi:10.1029/2017JD027970.

Vadas and Becker (2019): Numerical modeling of the generation of tertiary gravity waves in the mesosphere and thermosphere during strong mountain wave events over the Southern Andes, J. Geophys. Res. Space Phys., doi:10.1029/2019JA026694.

Becker et al. (2022): A high-resolution whole-atmosphere model with resolved gravity waves and specified large-scale dynamics in the troposphere and stratosphere, J. Geophys. Res. Atmos., doi:10.1029/2021JD035018.

**Wiley Editing Services**

**Message from your editor, Will**

Dear Author,

It was a pleasure working on your document. Do go through my changes and comments in the edited file.Please send me your feedback or any questions through your account (cn.wileyeditingservices.com).

**Editor's report**

I have provided feedback on your manuscript through specific comments along with ratings for relevant sections. The key below the table explains my ratings. I hope you find my feedback useful.

| Section | Rating |
|---|---|
| **Title** | ★ ★ |
| **Abstract** | ★ ★ |
| **Introduction** | ★ ★ ★ |
| **Data and methods** | ★ ★ ★ |
| **Results and discussion** | ★ ★ |
| **Summary** | ★ ★ |

★ ★ ★   This section required only a few revisions.
★ ★   Most parts of this section required revision.
★   The entire section required significant revision. Please go through my comments/changes carefully.

**Comments**

**NOVELTY OF THE STUDY**

The novelty could be more explicitly stated. E.g., This was the first study to… or This study for the first time…

**RELEVANCE AND CONTRIBUTION OF THE STUDY**

More discussion on the relevance and contribution of the study could be given in this paper. For example, the discussion should discuss the wider implications of these results, how they can be used. Discuss the potential shortcomings and limitations of the interpretations, their integration into the current understanding of CGWs and how this advances the current views.

[Figure]

**Wiley Editing Services**

**SUBMISSION READINESS**

No target journal was provided, therefore I edited the language and grammar and edited for general academic tone and writing conventions. Overall, the language was good but there are some areas that required heavier edits. Please see my comments for some specific areas that require your attention.

**Abstract**

The abstract should generally end with a line or two discussing the wider implications of the paper. What do these findings tell us? How can they be used?

**Introduction**

Provides a good level of background information and explicitly states the objectives.

**Data and Methods**

This section was well written and described the methods in sufficient detail. However, there were some methods discussed in the discussion section that were not first described here. For example, TIMED/SABER is not explained here but is discussed later on. This is not clear as the reader doesn't know what TIMED/SABER is.

**Results**

This was a detailed results section that made good use of figures. Please see my comments for some specific areas that require your attention.

**Discussion**

This is a good discussion of the data and brings in literature. However, in some areas it feels more like a results section in that data are being listed. More discussion could be given here. This section should discuss the wider implications of these results, how they can be used. Discuss the potential shortcomings and limitations of the interpretations, their integration into the current understanding of CGWs and how this advances the current views.

**Summary**

A good final summary section.

**Quick tip**

**Guideline**
Wordiness (the use of many words to convey an idea) should be avoided in academic writing.

**Explanation**

The use of too many words to convey one idea can muddle the message and divert the reader's attention. Therefore, in writing, especially academic writing, ideas need to be conveyed as concisely as possible. One way of doing this is to use concise alternatives to phrases. For example, the phrase "all over the world" can be replaced with the word "globally" or "worldwide."

Concise alternatives can also lend a more formal tone to the sentence. For example, "gradually" is considered a more formal alternative to "little by little" and is preferred in academic writing.

Finally, where possible, a direct verb (action) should be used instead of using a noun and verb.

[Figure]

Wiley Editing Services

For example, "segmentation of images was done" can be replaced with "images were segmented," which is clearer and preferred in academic writing.
* * *
**Example**

Before: We found that strong CGWs with clear signs of dissipation and/or nonlinearity were observed by the OH airglow network

After: The OH airglow network observed strong CGWs with clear signs of dissipation and/or nonlinearity

---

## Author Response (AR3)

**General comment:**

The revised manuscript has substantially improved compared to the former version. I believe that the science is in most parts well presented and that the general conclusions are convincing. Even though the language has improved, there are still numerous instances where the language or the writing is not sufficient and needs to be further improved. Only after this major editorial work is done, the paper is acceptable for publication in AC. Below I list most of the places that I found would need editorial work. I also include a few minor comments regarding the science.

Thank you very much for your good comments concerning our manuscript entitled "How do gravity waves triggered by a typhoon propagate from the troposphere to the upper atmosphere?". Those comments are all valuable and very helpful for revising and improving our paper, as well as the important guiding significance to our researches. We also thank you very much for your help in modifying the expression of English sentences. We have studied comments carefully and have made corrections which we hope meet with approval.

The detailed point-by-point responses are given below.

Minor comments:

L22: exponentially with height

**Response:**

Thank you very much for your suggestion. "exponentially" is replaced by "exponentially with height" in the revised manuscript. (Please see line 23 in the manuscript with track)

L25: mechanism of the [you do not show a mechanism that relates to typhoons in general: you mean a specific typhoon]

**Response:**

Thank you very much for your comment.

We re-described this sentence as: "We used ERA-5 reanalysis data and Multi-functional Transport Satellite-1R observations to quantitatively describe the propagation processes of typhoon-generated CGWs from the troposphere, through the stratosphere and mesosphere, to the thermosphere. " in the revised manuscript. (Please see line 25-28 in the manuscript with track)

L28: "like the relay" ???

Response:

"like the relay" is discarded from the revised manuscript. (Please see line 30 in the manuscript with track)

L32-38: This paragraph contains about a dozen typos.

**Response:**

Thank you very much for your comment. We re-described this paragraph as: "Gravity waves (GWs) can transfer momentum and energy from the lower to the upper atmosphere, thereby affecting global circulation and the thermal and compositional structures in the middle and upper atmospheres (Holton, 1983; Fritts and Alexander, 2003). Studies of dynamical, photochemical, and electrodynamics processes have indicated that GWs are fundamental for the coupling process between the troposphere, stratosphere, mesosphere, and thermosphere (Liu and Vadas, 2013; Smith et al., 2013; Vadas and Liu, 2013; Xu et al., 2015; Vadas and Becker, 2019)." in the revised manuscript. (Please see line 34-41 in the manuscript with track)

L39: of GWs and considered

**Response:**

"of GWs considered " is replaced by " of GWs and considered" in the revised manuscript. (Please see line 42 in the manuscript with track)

**L40-41: by GW breaking**

**Response:**

```
" primary wave " is replaced by " GW" in the revised manuscript.
(Please see line 43-44 in the manuscript with track)
```

L56-59: This somewhat confusing. First, it has to be "wave-wave interaction" and "wave-mean flow interaction". Second, wave-mean flow interaction is at the beginning of the mechanism for secondary GW generation discussed in the papers by Vadas and coauthors. Please reformulate accordingly.

**Response:**

Thank you very much for your comment.

We re-described this sentence as: "Moreover, wave-wave interaction, wave-mean flow interaction (Franke and Robinson, 1999; Vadas and Fritts, 2001), self-acceleration, and nonlinear breaking are other potential secondary wave generation mechanisms (Lund and Fritts, 2012; Fritts et al., 2015; Dong et al., 2020; Fritts et al., 2020; Zhou et al. 2002; Heale et al. 2020). " in the revised manuscript. (Please see line 62-66 in the manuscript with track)

L65: the lower to the upper atmosphere.

**Response:**

" the lower atmosphere to the upper atmosphere" is replaced by " the lower to the upper atmosphere" in the revised manuscript. (Please see line 70 in the manuscript with track)

L73-74: ".. was utilized to identify the mesosphere and thermosphere via ray tracing" – This does not make sense. Please reformulate.

**Response:**

Thank you very much for your comment.

We reformulate this sentence as:" However, given the observational limitations between the mesosphere and thermosphere, the two layers are connected by ray tracing theory. " in the revised manuscript.(Please see line 78-80 in the manuscript with track)

L74: to (a) investigate

**Response:**

"scrutinize" is changed to " investigate". (Please see line 82 in the manuscript with track)

L83-85 atmospheric waves in the middle and upper atmosphere triggered by severe events such as typhoons, earthquakes, and tsunamis.

**Response:**

We reformulate this sentence as:"The research aim of the DLAN is to explore the physical mechanism of vertical and horizontal propagation and the evolution of atmospheric waves in the middle and upper atmosphere triggered by severe disasters, such as typhoons, earthquakes, and tsunamis. " in the revised manuscript.(Please see line 90-93 in the manuscript with track)

L95-96: ... an excellent opportunity for studying coupling processes between the mesosphere and thermosphere.

**Response:**

We reformulate this sentence as: "Furthermore, the DLAN provides an excellent solution for studying the coupling processes between the mesosphere and thermosphere." in the revised manuscript.(Please see line 104-106 in the manuscript with track)

L123-125: discard "which provides ... layers"

**Response:**

As your suggestion, "which provides an excellent example for observing the CGWs stimulated by the typhoon and studying the coupling among the atmospheric layers" is discarded from the revised manuscript. (Please see line 134-135 in the manuscript with track)

L141: mention the cadence of the ERA-5 data

**Response:**

We reformulate this sentence as: "Horizontal reanalysis temperature and wind data with a pre-interpolated resolution of  $0.25^{\circ} \times 0.25^{\circ}$  and time resolution of 1 h were used in this study." in the revised manuscript.(Please see line 151-152 in the manuscript with track)

L157: How is \omega related to \omega\_{Ir} in Eq (1)?

**Response:**

 $\omega_{lr} = \omega_r - (ku + lv)$ , where  $\omega_{lr}$  is the intrinsic frequency,  $\omega_r$  is ground-based frequency. (Please see line 158 in the manuscript with track)

L168-169: We extracted the stratospheric CGW excited by the typhoon from ERA-5 reanalysis.

**Response:**

We reformulate this sentence as: "We extracted the stratospheric CGW excited by the typhoon from ERA-5 reanalysis" in the revised manuscript. (Please see line 185-186 in the manuscript with track)

L178-179: ... temperature perturbation at 60 km in ERA-5 was only ...

**Response:**

Thank you very much for your comment. "altitude" is replaced by "in ERA-5" in the revised manuscript. (Please see line 196 in the manuscript with track)

L183-185: This citation of Liu et al. (2014) does not seem to fit perfectly well here. You want to cite a paper that shows that the larger-scale CGW arrive earlier at higher altitudes (have faster vertical group velocities) than the smaller-scale waves. This mechanism was discussed in detail by Vadas and Azeem (2021, JGR-SP), https://doi.org/10.1029/2020JA028275

**Response:**

Thank you very much for your comment and suggestion.

Reference Vadas and Azeem (2021) is appropriately cited here, while reference Liu et al. (2014) is moved to the Introduction section. (Please see line 206-208 and line 49-51 in the manuscript with track)

L207-208: This sounds very confusing. Perhaps the 3rd author can reformulate this?

**Response:**

We reformulate this sentence as:" As long as the CGWs do not encounter the critical layer or break, the CGWs generated in the lower atmosphere can propagate to the OH airglow layer." in the revised manuscript.(Please see line 228-230 in the manuscript with track)

L209-210: I do not understand: You see different predominant horizontal wavelengths at 20 and 40 km!

**Response:**

We reformulate this sentence as:"A single dominant horizontal wavelength is seen at each altitude of 20 km, 40 km, and 60 km in the ERA-5 reanalysis. " in the revised manuscript. (Please see line 231-232 in the manuscript with track)

L211: "due to the limited resolution" – what is the context here??? Note that the 150 and 300 km wavelengths that you find are very well resolved by ERA-5.

**Response:**

"due to the limited resolution" is discarded from the revised manuscript. (Please see line 232-233 in the manuscript with track)

213: discard "as the imager ... resolution". I have no idea why and how you want to relate the resolution of the imager to the resolution of ERA-5.

**Response:**

"as the imager has much higher spatial resolution" is discarded from the revised manuscript. (Please see line 234-235 in the manuscript with track)

L215-216: very confusing formulation. Third author, please step in. What is a "phase plane"???

**Response:**

We are very sorry to confuse you. We realized that this description was inaccurate, so we reformulate this sentence as:"To verify whether the same wave was propagated from the reanalysis data layer to the OH layer, we used the group velocity to estimate the time when the

CGW at the altitudes of 20 km, 40 km, and 60 km reached the OH airglow layer. " in the revised manuscript. (Please see line 237-240 in the manuscript with track)

L220-222: Therefore, the times when the CGWs visible in ERA-5 at 60 km, 40 km, and 20 km would reach the OH airglow layer are approximately ...LT, .... LT, and ... LT, ...

**Response:**

We reformulate this sentence as: "Therefore, the times when the CGWs visible in ERA-5 at 60 km, 40 km, and 20 km would reach the OH airglow layer are approximately 23:21 LT, 23:36 LT, and 23:53 LT as shown in Fig. 5a, 5b, and 5c, respectively. " in the revised manuscript. (Please see line 242-245 in the manuscript with track)

L229-231: ... tracked over different altitudes and that the CGWs in the mesosphere propagated upward from the stratosphere. [You did not show in this paper that the GWs you see in ERA-5 were excited by the typhoon. This is just a reasonable assumption that you made.]

**Response:**

Thank you very much for your comment. We reformulate this sentence as:" This suggests that the same CGW event can be perfectly tracked over different altitudes and that the CGWs in the mesosphere propagated upward from the stratosphere." in the revised manuscript. (Please see line 252-255 in the manuscript with track)

L242: "The observation period" ??? Do you mean "observed wave period" ?

**Response:**

Yes, you are right.

"observation period" is changed to "observed wave period". (Please see line 266 in the manuscript with track)

L243: why "multi-scale"? You are discussing predominant wave parameters, or not?

**Response:**

Thank you very much for your comment.

"multi-scale" is discarded from the revised manuscript. (Please see line 268 in the manuscript with track)

L246: "faster phase speed and shorter period" than what?

**Response:**

Thank you very much for your comment. We reformulate this sentence as:" The CGW observed in the OI 630.0 nm airglow had much faster phase speed and shorter period than that observed

in the mesosphere " in the revised manuscript. (Please see line 270-271 in the manuscript with track)

L248-250: "Indeed ... km." – This sentence is hard to understand. Please reformulate.

**Response:**

Thank you very much for your comment. We reformulate this sentence as:" Indeed, compared with the long-distance extension of the CGWs in the mesosphere, the horizontal propagation distance of the CGWs in the thermosphere was only 600 km from OI 630.0 nm network observation." in the revised manuscript. (Please see line 273-276 in the manuscript with track)

L250-252: Vadas and Crowley (2010) showed that thermospheric GWs may be secondary GWs generated by the breaking of primary GWs in the mesosphere and thermosphere.

**Response:**

Thank you very much for your comment. We reformulate this sentence as:"Vadas and Crowley (2010) showed that thermospheric GWs may be secondary GWs generated by the breaking of primary GWs in the mesosphere and thermosphere. " in the revised manuscript. (Please see line 278-280 in the manuscript with track)

L252-253: "We argue .... Pattern." – This is sentence does not make sense.

**Response:**

Thank you very much for your comment. We reformulate this sentence as:"We argue that the thermospheric CGW observed by the OI 630.0 nm airglow imager was not directly generated by the typhoon, but a secondary GW. " in the revised manuscript. (Please see line 280-283 in the manuscript with track)

L269: We started the ray-tracing at heights of 240 km, 250 km, and 260 km , and analyzed the results.

**Response:**

Thank you very much for your comment. We reformulate this sentence as:" We started the ray-tracing at heights of 240 km, 250 km, and 260 km, and analyzed the results. " in the revised manuscript. (Please see line 296-298 in the manuscript with track)

L274: that a reflection layer was encountered. According to linear theory, this suggests that ...

Response:

Thank you very much for your comment. We reformulate this sentence as:"Subsequently, seven backward traced trajectories took 37 minutes and terminated at an altitude of approximately 95 km thereby indicating that a reflection layer was encountered. According to linear theory, this suggests that the thermospheric CGW could not have come from below 95 km. " in the revised manuscript. (Please see line 301-305 in the manuscript with track)

L275: ... 95 km. Therefore, the thermospheric GWs must have ...

**Response:**

Thank you very much for your comment. We reformulate this sentence as:"The thermospheric GW must have been generated at any altitude between 95 km and the altitude of the OI 630.0 nm airglow. " in the revised manuscript. (Please see line 305-306 in the manuscript with track)

L278: ... 00:23 LT. Figure 9 ...

**Response:**

"Meanwhile," is discarded from the revised manuscript. (Please see line 308 in the manuscript with track)

L286-287: Please discard "which showing clear signs ... processes"! Also note that you mention here the same conclusions that you already stated 12 lines earlier.

**Response:**

Thank you very much for your suggestion.

", which showing clear signs of dissipation and/or nonlinear processes" is discarded from the revised manuscript. (Please see line 316 in the manuscript with track)

L291-305: These arguments are not conclusive. My understanding is that the reason for the difference between center of the fitting circle and the end points of the ray racing is not the fact that there are some large horizontal background winds. Rather, the background winds you used in the ray tracing model were presumably not realistic enough. This would be plausible because the HWM model is an empirical model. Another possible mechanism is that the wave phase speeds are accelerated by accelerating background winds. Does your ray tracing model include this transient effect (time derivatives of the background wind components giving rise to time derivative of the frequency for a particular ray)? You should include the answer to this question in your Sec. 2.4.

**Response:**

We appreciate your constructive suggestions, which are very beneficial for improving our paper. According to your suggestion, the following discussions are added to the revised manuscript. "In this study, we assume that the background wind field is independent of time, so ground-based frequency  $\omega_r$  remains constant along a ray's path (Lighthill, 1978). However, the actual wind field changes with time, which may lead to deviation between the ray tracing results and the wave source locations." (Please see line 171-175 in the manuscript with track)

"Another possible mechanism is that the wave phase speeds are accelerated by accelerating background winds. As mentioned above, the ground-based frequency  $\omega_r$  remains constant along a ray's path assuming the background wind field is independent of time (Lighthill, 1978). However, transient effect (time derivatives of the background wind components giving rise to time derivative of the frequency for a particular ray) may cause the phase speeds to be accelerated, which may lead to the ray-tracing results did not match the real locations." (Please see line 333-339 in the manuscript with track)

L331-332: This sentence cannot relate to what has been shown in this paper up until here. Please discard this sentence and start the paragraph with "Figure 11 shows ...".

**Response:**

Thank you very much for your suggestion.

"We elucidated the dissipation process of the CGWs in detail by examining the evolution process of their amplitude." is discarded from the revised manuscript. (Please see line 368-369 in the manuscript with track)

L333: Which wave fronts???

**Response:**

"wavefronts" is replaced by "wavefronts (w1-w5)". (Please see line 370 in the manuscript with track)

L33-334: "A dominant ... confirmed." – Sorry, this cannot be concluded based on Fig. 11. Please reformulate.

**Response:**

We are very sorry for not showing it clearly. We have re marked it in the Fig. 11. Please check.

L334: "As a result" ??? What logical connection do you have in mind here?

**Response:**

"As a result" is discarded from the revised manuscript. (Please see line 371 in the manuscript with track) L343-346: Sorry, but I cannot follow these formulations. Please reformulate.

**Response:**

Thank you very much for your comment. We reformulate this sentence as:" We obtained similar scale fluctuations were obtained in the two airglow layers. The horizontal wavelength of the wave obtained by the OI 630 nm airglow layer was approximately 118 km. The OH airglow layer has also obtained near-scale fluctuations with wavelengths of approximately 109 km. These waves could be the same waves seen in the thermosphere. " in the revised manuscript. (Please see line 380-384 in the manuscript with track)

L357: Note that wave amplitude fluctuations can ...

**Response:**

We reformulate this sentence as:"Note that wave amplitude fluctuations can also result from the transient nature of the wavepacket. " in the revised manuscript. (Please see line 395-396 in the manuscript with track)

L358-360: Please reformulate!

**Response:**

We reformulate this sentence as:" The propagation state can be studied by using the dispersion relationship with GW. However, the dissipation region of the CGW lacks the real-time background temperature and wind field." in the revised manuscript. (Please see line 396-398 in the manuscript with track)

L360: TIMES/SABER etc. appears out of the blue. Please reformulate and provide the context.

**Response:**

"TIMES/SABER" is changed to "the limb-viewing of Sounding of the Atmosphere using Broadband Emission Radiometry (SABER) instrument on the Thermosphere Ionosphere Mesosphere Energetics and Dynamics (TIMED) satellite". (Please see line 398-401 in the manuscript with track)

L361-362: "On this basis" ??? What is the logic here?

**Response:**

We reformulate this sentence as:" Background wind field data were obtained from an ATRAD MDR6 all-sky VHF meteor radar at Beijing station. " in the revised manuscript. (Please see line 402-405 in the manuscript with track)

L366: "sound"???

Response:

"sound" is changed to " measurement location ". (Please see line 408 in the manuscript with track)

Figure 13, caption: Which color is what?

Response:

We reformulate the caption as below:

Figure 13. Vertical wave number  $m^2$  profile (black) derived from the temperature from TIMED/SABER measurement location at 04:18:49 LT and the meteor radar wind from Beijing station marked in Fig. 9. The red line represents the OH1.6  $\mu$ m emission intensity obtained by the TIMED/SABER. The horizontal blue lines represent the top and bottom boundaries of the duct region.

(Please see the caption of Fig. 13 in the manuscript with track)

L402-403: ... to connect GWs in the upper mesosphere to GWs in the thermosphere at about 250.

**Response:**

We reformulate this sentence as:" Due to the observational limitations, a backward ray-tracing theory was used to connect GWs in the upper mesosphere to GWs in the thermosphere at about 250 km." in the revised manuscript. (Please see line 445-447 in the manuscript with track)

L404-405: The fact that your rays terminated in the upper mesosphere does NOT imply "clear signs of primary GW dissipation and/or nonlinear processes". Please discard this phrase.

**Response:**

Thank you very much for your suggestion.

", which shows clear signs of primary CGW dissipation and/or nonlinear processes" is discarded from the revised manuscript. (Please see line 448-449 in the manuscript with track)

L406: the OH network

**Response:**

" OH network " is replaced by " the OH network" in the revised manuscript. (Please see line 450 in the manuscript with track)